# From Natural Alignment to Conditional Controllability in Multimodal Dialogue

**Zeyu Jin**[1,*], **Songtao Zhou**[1,*], **Haoyu Wang**[1], **Minghao Tian**[2], **Kaifeng Yun**[3], **Zhuo Chen**[4], **Xiaoyu Qin**[1,†], **Jia Jia**[1,5,†]

[1]Department of Computer Science and Technology, Tsinghua University, [2]Rice University, [3]Tsinghua Shenzhen International Graduate School, Tsinghua University, [4]ByteDance, [5]BNRist, Tsinghua University
{jinzeyu23, zst24}@mails.tsinghua.edu.cn
{xyqin, jjia}@tsinghua.edu.cn

## Abstract

The recent advancement of Artificial Intelligence Generated Content (AIGC) has led to significant strides in modeling human interaction, particularly in the context of multimodal dialogue. While current methods impressively generate *realistic* dialogue in isolated modalities like speech or vision, challenges remain in *controllable* Multimodal Dialogue Generation (MDG). This paper focuses on the natural alignment between speech, vision, and text in human interaction, aiming for expressive dialogue generation through multimodal conditional control. To address the insufficient richness and diversity of dialogue expressiveness in existing datasets, we introduce a novel multimodal dialogue annotation pipeline to curate dialogues from movies and TV series with fine-grained annotations in interactional characteristics. The resulting MM-Dia dataset (360+ hours, 54,700 dialogues) facilitates *explicitly* controlled MDG, specifically through style-controllable dialogue speech synthesis. In parallel, MM-Dia-Bench (309 highly expressive dialogues with visible single-/dual-speaker scenes) serves as a rigorous testbed for *implicit* cross-modal MDG control, evaluating audio-visual style consistency across modalities. Extensive experiments demonstrate that training on MM-Dia significantly enhances fine-grained controllability, while evaluations on MM-Dia-Bench reveal limitations in current frameworks to replicate the nuanced expressiveness of human interaction. These findings provides new insights and challenges for multimodal conditional dialogue generation. [1]

## 1 Introduction

Dialogue has long been considered one of the most natural forms of human interaction, involving multiple communication channels such as text, speech, vision, gestures, and etc. In the AIGC era, multimodal dialogue has become increasingly important for a wide range of applications in *human-computer interaction*, *social computing*, and *film-making*.

Existing research in dialogue primarily falls into two directions: (1) Semantic generation, which emphasizes producing coherent and contextually appropriate responses, as exemplified by large-scale dialogue systems like ChatGPT (OpenAI et al., 2024). (2) Modality mapping, which projects the given semantics into output modalities such as speech (Zhu et al., 2025; Zhang et al., 2024) and motion (Kong et al., 2025b). However, both directions prioritize the realistic transmission of modality-isolated dialogue content, while neglecting the systematic modeling of cross-modal interactional style. This results in limited expressiveness and controllability in the generated outputs.

Controllable multimodal dialogue generation presents several major challenges: (1) **Lack of high-quality native multimodal dialogue data.** Existing multimodal dialogue datasets, as shown in

---

[*]Equal contribution
[†]Corresponding author

[1]https://github.com/jessyjinzy/MM-Dia

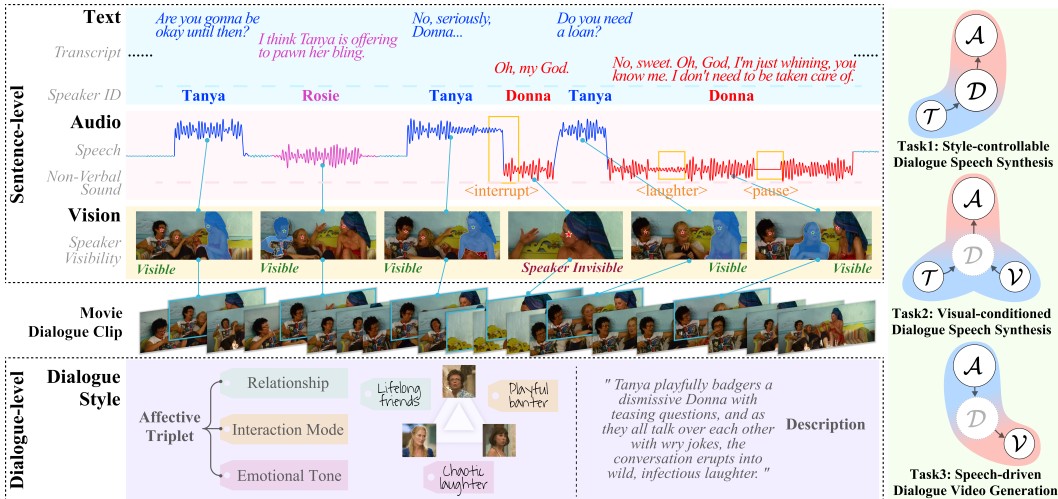

Figure 1: An example dialogue clip from MM-DIA and MM-DIA-BENCH with hierarchical (sentence- and dialogue-level) annotations, featuring rich multimodal dialogue interaction details. The right panel depicts three multimodal dialogue generation tasks involving text ($\mathcal{T}$), audio ($\mathcal{A}$), vision ($\mathcal{V}$) and dialogue style ($\mathcal{D}$), demonstrating both explicit (*Task 1*) & implicit control (*Task 2,3*).

Tab. 16, suffer from narrow sources and missing modalities, which restricts their expressiveness and ability to portray complex multimodal interactions. This is largely due to the difficulty of collecting synchronized text-audio-visual dialogue data. (2) **Lack of scalable annotation methods for interaction-level semantics.** Human categorically labeled datasets like MELD (Poria et al., 2019) and MC-EIU (Liu et al., 2024b) fail to capture the nuanced, continuous nature of human interaction, while remaining costly and difficult to scale. (3) **Lack of systematic benchmarks and evaluation protocols.** Current dialogue benchmarks (Poria et al., 2019; Liu et al., 2024b) primarily focus on local semantic coherence and modality-isolated fidelity, leaving a methodological gap in evaluating dialogue-level controllability and cross-modal style consistency.

In this paper, we address these challenges by constructing a large-scale expressive multimodal dialogue dataset, introducing new annotation paradigms, and establishing systematic benchmarks for controllable multimodal dialogue generation.

Addressing the data scarcity, we develop an automatic data pipeline to curate synchronized multimodal dialogues and generate fine-grained interaction-level annotations from in-the-wild movies and TV series. To overcome the complex scene transitions and audio-visual asynchrony inherent in cinematic data, we propose specialized techniques for robust dialogue boundary segmentation and multimodal speaker identification. To facilitate versatile controllability across diverse scenarios, we formulate two complementary paradigms of *dialogue expressiveness*: (1) **Affective Triplet**, a structured schema of ⟨*Relationship*, *Interaction Mode*, and *Emotional Tone*⟩ to jointly encapsulate role identity, social interplay, and emotion dynamics; and (2) **Freestyle Description**, capturing per-speaker, turn-level style trajectories. Extensive validation demonstrates that our pipeline achieves human-level quality in annotation consistency and reliability.

Applying the proposed pipeline to over 700 hours of movies and TV series, we present MM-DIA, a multimodal dialogue dataset characterized by 360.26 hours, 54,700 clips of highly expressive, contextually rich, and interaction-heavy dialogues. MM-DIA features fine-grained annotation across various dialogue dimensions, including non-verbal sound, speaker identity and emotional dynamics at both the individual (sentence-wide) and collective (dialogue-wide) levels. To our best knowledge, it is the first dataset to specifically center on dialogue expressiveness across multiple modalities.

Leveraging MM-DIA, we formally introduce Multimodal Dialogue Generation (MDG) as a conditional generation paradigm. Given a multimodal conversational context (text, audio, vision), MDG task aims to synthesize multimodal dialogues that maintain rigorous cross-modal alignment while enabling fine-grained controllability over interaction-level variables. To operationalize this, we categorize the conditional control into *explicit control*, where styles are specified via natural language prompts, and *implicit control*, where conditions are inferred via cross-modal cues.

Table 1: Comparison of the MM-DIA with dialogue-related datasets across domain, scale, modality, annotation, and *open-source* (OS). Modality includes *text* ($\mathcal{T}$), *vision* ($\mathcal{V}$), and *audio* ($\mathcal{A}$), with audio-visual details on *speaker identity* (S-ID), *non-verbal cues* (N-V), and *speaker visibility* (S-V).

| Domain | Dataset | Scale | | | Modality | | | Audio-visual Details | | | Annotation | | OS |
|---|---|---|---|---|---|---|---|---|---|---|---|---|---|
| | | #Clip | #Utt. | #Dur.(h) | $\mathcal{T}$ | $\mathcal{V}$ | $\mathcal{A}$ | S-ID | N-V | S-V | Granularity | Label | |
| Spoken Dia. | OpenDialogue (Zhu et al., 2025) | 1M | 6.5M | 6.8K | ✓ | ✗ | ✓ | ✓ | ✗ | ✗ | Dialogue | None | ✓ |
| Textual Dia. | OpenViDial 2.0 (Wang et al., 2021) | - | 5.6M | - | ✓ | ✓ | ✗ | ✗ | ✗ | ✗ | Dialogue | None | ✓ |
| | YTD-18M (Han et al., 2023) | 18M | - | - | ✓ | ✓ | ✗ | ✗ | ✗ | ✓ | Dialogue | None | ✓ |
| Text-to-Video | OpenVid-1M (Nan et al., 2025) | 1M | - | 2.1K | ✓ | ✓ | ✗ | ✗ | ✗ | ✗ | Scene | Desc. | ✓ |
| | Captain Cinema (Xiao et al., 2026) | - | 300K | 500.0 | ✓ | ✓ | ✗ | ✓ | ✗ | ✗ | Shot | Desc. | ✗ |
| MM Dia. Und. | MELD (Poria et al., 2019) | 1.4K | 14K | 13.6 | ✓ | ✓ | ✓ | ✓ | ✗ | ✓ | Sentence | Tag | ✓ |
| | MC-EIU (Liu et al., 2024b) | 5.0K | 56K | 53.0 | ✓ | ✓ | ✓ | ✓ | ✗ | ✓ | Sentence | Tag | ✓ |
| Movie Gen. | MovieBench (Wu et al., 2025) | 16.0K | 61K | 69.2 | ✓ | ✓ | ✓ | ✓ | ✗ | ✗ | Shot/Scene | Desc. | ✓ |
| **MM Dia. Gen.** | **MM-DIA (Ours)** | **54.7K** | **449K** | **360.3** | ✓ | ✓ | ✓ | ✓ | ✓ | ✓ | **Dia./Sent.** | **Desc./Tag** | ✓ |
| **MM Dia. Gen.** | **MM-DIA-BENCH (Ours)** | **309** | **1,851** | **1.7** | ✓ | ✓ | ✓ | ✓ | ✓ | ✓ | **Dia./Sent.** | **Desc./Tag** | ✓ |

For the explicit prompt control, we introduce **Style-controllable Dialogue Speech Synthesis** (Fig. 1 *Task 1*) to generate conversational audio consistent with the freestyle descriptions. Through supervised finetuning on MM-DIA, existing models produce high-quality outputs with enhanced intelligibility and turn-taking accuracy, while faithfully reflecting the instructed styles.

For the implicit cross-modal control, we introduce two tasks: (1) **Vision-conditioned Dialogue Speech** (Fig. 1 *Task 2*) which synthesizes contextually accurate speech aligned with turn-taking visual sequences of a dialogue; (2) **Speech-driven Dialogue Video Generation** (Fig. 1 *Task 3*), which generates expressive video from conversational audio. Both tasks demand modeling implicit alignment in cross-modal interactions, which is difficult to quantify. To facilitate standardized evaluation, we build MM-DIA-BENCH, a diverse and balanced benchmark of 309 highly expressive dual-speaker dialogues with ensured speaker visibility. The benchmark evaluates audio-visual style consistency across dialogue turns, addressing a limitation of existing video evaluation protocols that overlook cross-modal stylistic alignment. Experiments results highlight the limitations of current frameworks in maintaining audio-visual consistency when replicating the expressiveness of human interaction, offering new insights and challenges in multimodal dialogue generation.

## 2 RELATED WORKS

### 2.1 MULTIMODAL DIALOGUE DATASETS

Multimodal dialogue datasets have driven recent progress in multimodal AI systems. Large scale spoken dialogue corpora (Zhu et al., 2025; Jin et al., 2024) support speech-based systems but remain limited to unimodal settings, hindering multimodal alignment and cross-modal interaction.

Table 2: Comparison between MM-DIA and existing TV/Movie-sourced datasets in the annotation framework.

| Dataset | Source | Segmentation | Anno. Input | Anno. Tool |
|---|---|---|---|---|
| MELD (Poria et al., 2019) | TV | Human | $\mathcal{V} + \mathcal{A} + \mathcal{T}$ | Human |
| MC-EIU (Liu et al., 2024b) | TV | Human | $\mathcal{V} + \mathcal{A} + \mathcal{T}$ | Human |
| MovieBench (Wu et al., 2025) | Movie | Vision-based | $\mathcal{I} + \mathcal{A} + \mathcal{T}$ | GPT-4o |
| **MM-DIA (Ours)** | **TV/Mov.** | **Multimodal** | $\mathcal{V} + \mathcal{I} + \mathcal{A} + \mathcal{T}$ | **Gemini 2.5-pro** |

Web sourced video datasets (Ju et al., 2024; Nan et al., 2025) offer abundant audio-visual data, yet mostly consist of casual chitchat or predefined scenarios, limiting diversity for prompt-controlled dialogue generation. Movie-sourced video datasets (Han et al., 2024; Wu et al., 2025) offer expressive audio-visual content but typically lack clear dialogue boundary delineation. To address these limitations, we introduce a novel multimodal framework catering for dialogue-level style annotation, as shown in Tab. 2. Unlike prior approaches that rely on isolated key-frame image sequences ($\mathcal{I}$), we leverage synchronized audio-visual inputs to enable versatile datasets with fine-grained style annotations for advancing multimodal dialogue modeling.

### 2.2 DIALOGUE GENERATION FROM MULTIMODALITY

Dialogue serves as *the smallest and most structured unit* of human interaction. Recent progress in spoken dialogue generation (Nari Labs, 2025; Ju et al.; Zhu et al., 2025) has captured realistic turn-

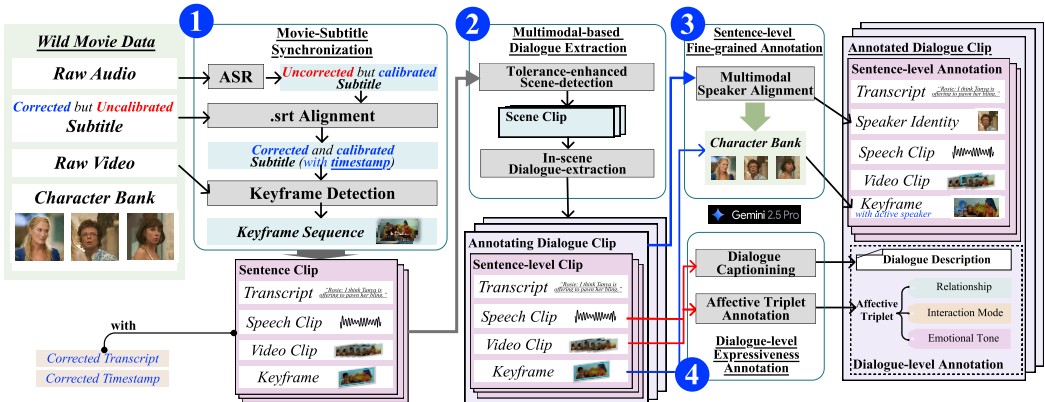

Figure 2: Framework of the Movie/TV-sourced in-the-wild data curation pipeline for multimodal dialogue extraction with fine-grained interaction-level annotations.

taking and multi-speaker timbres (Boson AI, 2025; Zhou et al., 2024), enabling more natural verbal exchanges. In parallel, cinematic video generation (Xiao et al., 2026; Liu et al., 2024a; Blattmann et al., 2023) supports high-fidelity multi-shot scenes with consistent character appearances and immersive transitions, producing coherent visual narratives. Specialized systems further bridge these modalities through synchronized talking head generation (Cui et al., 2025; Wang et al., 2026) and vision-context-aware speech synthesis(Zhou et al., 2025b; Wang et al., 2025) to ensure expressive voice with coherence visual appearance. Although these advances establish the modality-specific foundations for conveying semantic information, achieving fine-grained cross-modal control across speech, vision, and text for expressive multi-speaker dialogue remains an open challenge. In this paper, we seek to address this gap by introducing the necessary infrastructure (dataset, pipeline, benchmark, and task formulation) for controllable multimodal dialogue generation.

## 3 MM-DIA: A LARGE-SCALE EXPRESSIVE MULTIMODAL DIALOGUE DATASET

Movies and TV series are two of the richest artistic forms that feature carefully crafted, context-sensitive performances, but their complex audio-visual asynchrony poses significant challenges in data processing. In this section, we describe our pipeline for movie/TV-sourced in-the-wild dialogue extraction and fine-grained interaction-level annotations, followed by a statistical analysis and validation of the resulting data.

### 3.1 PIPELINE ORIENTATION WITH DATA PREPARATION

Cinematic dialogues exhibit stronger emotion, heightened tension, and greater resemblance to everyday interactions. However, the pursuit of strong sensory effects introduce substantial noise. Frequent background sounds, dramatic bursts, or ambiguous murmurs hinder the accuracy of Automatic Speech Recognition (ASR), while artistic camera movements create voiceover and flashback, complicating dialogue boundary detection and speaker attribution. These factors call for a cautious and comprehensive multimodal processing strategy.

In preparation of the dataset, we collect raw movie & TV data from public available sources, noting that official subtitle (.srt) files are often unavailable. Although ASR can generate time-stamped transcriptions, its high word error rates limit reliability. To ensure high-quality subtitles and further enlarge the dataset while balancing temporal accuracy and textual fidelity, we additionally adopt multi-sourced uncalibrated subtitles and align them with ASR outputs (Fig. 2 *Step 1*, details in Appendix A.1). With the corrected timestamps from the calibrated subtitles, we extract a keyframe sequence from each subtitle line as representatives for dialogue boundary detection.

## 3.2 Multimodal-based Dialogue Extraction

Dialogue boundaries differ from shot or scene boundaries, as conversations may span multiple shots and topic shifts within a single long scene. To automatically extract continuous dialogue from movies & TVs, we introduce a tolerance-enhanced scene boundary detection method. It first applies a Vision-Language Model (VLM) to identify scene continuity, followed by a Large Language Model (LLM) to refine in-scene dialogue boundaries. Unlike traditional frame-to-frame matching methods (Wu et al., 2025; Xiao et al., 2026), our approach incorporates a buffer mechanism with a dynamic keyframe pool (Fig. 2 *Step 2*, details in Appendix A.2), allowing the model to bridge momentary visual disruptions such as rapid camera shifts, flashbacks, or perspective changes. This improves robustness in maintaining dialogue continuity across complex scenes. Based on the resulting scene-level segmentation, we further leverage subtitles and LLM-based semantic filtering to extract meaningful dialogue segments, particularly in long scenes exceeding 90 seconds. By combining visual and textual cues, the framework achieves coherent and accurate dialogue extraction, ensuring the integrity of multimodal context.

## 3.3 Sentence-level Fine-grained Annotation

Based on the identified boundaries, we segment the content into short dialogue clips. Next, we perform speaker attribution to each dialogue segment. However, conventional automatic tools cannot reliably perform speaker attribution in this setting. Audio-based speaker diarization often suffers from limited accuracy due to noisy acoustic conditions, while visual-based active speaker detection is unreliable because speakers are frequently not visible in cinematic content. To address this issue, we employ Gemini-2.5-flash to perform speaker attribution based on synchronized audio-visual segments and corresponding subtitles (Fig. 2 *Step 3*). The model is prompted with a predefined main character bank to recognize speakers; otherwise, it assigns speaker identities based on their on-screen personas. Additionally, we annotate non-verbal sounds and vocalizations to capture fine-grained expressive cues and contextual nuances in dialogue. To support downstream dialogue-related tasks such as talking head generation, we further use the Insightface toolkit to annotate the visibility of main speakers in the aligned keyframes.

## 3.4 Dialogue-level Expressiveness Annotation

To systematically study complex interaction-level behaviors, we define *dialogue expressiveness* as cross-modal consistency in interaction that extends beyond semantic content. We propose two complementary paradigms for modeling dialogue expressiveness:

(1) **Affective Triplet Control**, ⟨*Relationship, Interaction Mode, Emotional Tone*⟩, which jointly model role identity, social interplay, and emotional dynamics. It enables precise control over scenario-level dialogue behavior.

(2) **Description Control**, which captures per-speaker, turn-level style trajectories. It enables

Table 3: Detailed statistics for MM-DIA and MM-DIA-BENCH. Scored from Gemini/Human.

| Statistic | MM-DIA | MM-DIA-BENCH |
|---|---|---|
| Total Dialogues | 54,700 | 309 |
| Total Turns | 449,138 | 1,851 |
| Total Duration (h) | 360.26 | 1.69 |
| Avg. Spk. / Dia. | 2.29 | **2.00** |
| Avg. Dur. / Dia. (s) | 23.71 | 19.69 |
| Avg. Turns / Dia. | 8.21 | 5.99 |
| Avg. Dur. / Turn (s) | 2.89 | 3.29 |
| Avg. Turns / Spk. / Dia. | 3.59 | 3.00 |
| Avg. Rounds of Speaker Changes / Dia. | 4.28 | 4.09 |
| Speaker Visibility | Partial | **All** |
| Avg. Score on Emotion Intensity | 6.76 / 5.22 | **7.81 / 5.74** |
| Avg. Score on Volatility of Emotion Flow | 5.32 / 4.36 | **7.45 / 5.68** |

the separate control over speakers, and the fine-grained modeling of emotion flow across turns within the same speaker.

Together, these paradigms encompass both structured tag-based control and freestyle description-based natural language control. Given the speakers bank and synchronized audio-visual segments, we use Gemini-2.5-pro to annotate both paradigms of the dialogue expressiveness (Fig. 2 *Step 4*). Beyond qualitative annotation, we further introduce two complementary quantitative dimensions to measure dialogue expressiveness: global emotional intensity at the dialogue level and local emotional volatility at the speaker level. For instance, a consistently high-energy dialogue would be rated as high in emotional intensity but low in emotional volatility.

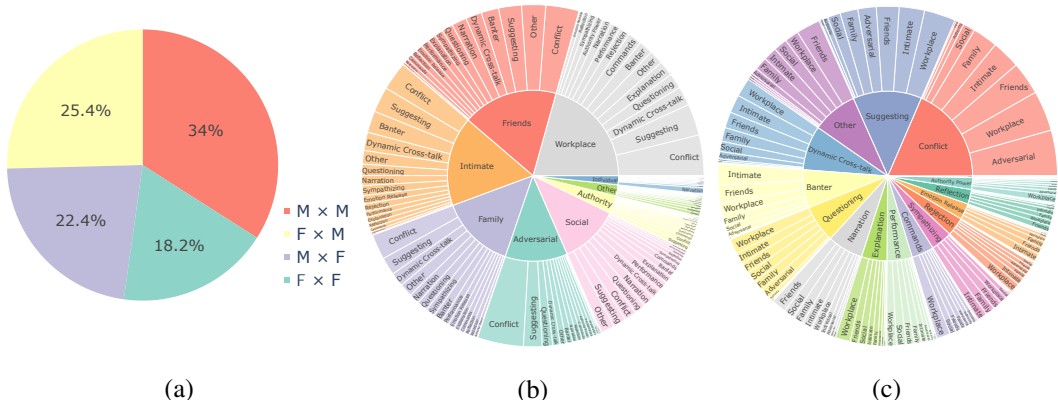

Figure 3: Distributions of (a) Dual-speaker Gender, (b) Relationship, and (c) Interaction Mode in MM-DIA.

### 3.5 MM-DIA WITH MM-DIA-BENCH

Applying our data curation pipeline to over 700 hours of raw content—including more than 200 movies and 9 TV series—we construct MM-DIA, a multimodal dialogue dataset comprising 360.26 hours, 54,700 clips of expressive, context-rich, and interaction-intensive dialogues. The dataset is accompanied by fine-grained annotations covering various dialogue aspects, such as non-verbal sounds, speaker identity and emotional dynamics at both individual and dialogue levels. To our knowledge, MM-DIA is the first dataset specifically centered on dialogue-level expressiveness across modalities. As illustrated in Fig. 3, MM-DIA exhibits a balanced distribution across affective triplet categories. Notably, exploratory analysis reveals meaningful associations between Relationships and Interaction Mode. For instance, *Commands* and *Questioning* are predominantly linked to *Workplace* settings, while *Intimate* relationships tend to involve *Emotion Release* and *Banter*. Such patterns provide qualitative evidence of distributional consistency between the MM-DIA and real-world social interactions.

Subsequently, we construct MM-DIA-BENCH, a diverse and balanced benchmark comprising 309 carefully selected instances of highly-expressive dual-speaker dialogues with ensured speaker visibility. The benchmark is designed to support the evaluations of various downstream tasks in cross-modal dialogue generation. As shown in Tab. 3, both Gemini model and human raters consistently assign higher expressiveness scores to MM-DIA-BENCH, validating its role as a high-expressiveness benchmark.

### 3.6 VALIDATION OF THE ANNOTATION SYSTEM AND DATASET

To validate the quality of the annotation system, we conduct a series of through evaluation (see Appendix A.3) on each step component, demonstrating that our pipeline achieves human-level quality in annotation consistency and reliability.

## 4 MULTIMODAL DIALOGUE GENERATION TASKS

In this section, we first introduce a unified formulation of ***Multimodal Dialogue Generation*** **(MDG)**. Subsequently, we present three representative tasks under different control conditions and output modalities, providing instantiations of the MDG framework.

### 4.1 PROBLEM FORMULATION

To enable systematic study of multimodal dialogue behaviors, we formalize MDG as a conditional generation problem. Given a multimodal context $\mathcal{C} = \{C_{\text{txt}}, C_{\text{aud}}, C_{\text{vis}}\}$, the goal is to generate multimodal dialogue behaviors $\mathcal{Y} = \{Y_{\text{txt}}, Y_{\text{aud}}, Y_{\text{vis}}\}$ that are (i) *semantically coherent*,(ii) *aligned across modalities*, and (iii) *controllable*. Formally, MDG can be expressed as modeling a conditional distribution: $P(\mathcal{Y} \mid \mathcal{C}, \mathcal{Z})$, where $\mathcal{Z}$ denotes explicit/implicit control variables for dialogue

style. This formulation unifies diverse downstream tasks such as style-controllable dialogue speech synthesis, vision-conditioned speech synthesis, and speech-driven dialogue video generation, providing a foundation for systematic benchmarking of controllable multimodal dialogue.

## 4.2 TASK 1: STYLE-CONTROLLABLE DIALOGUE SPEECH SYNTHESIS

**Definition.** Conditioning purely on text, this task synthesizes multi-speaker dialogue audio stream $A$ from a transcript $T = \{u_1, \ldots, u_N\}$ (consisting of $N$ utterances) and an explicit style condition $Z_{\text{exp}}$. The condition $Z_{\text{exp}}$ belongs to either a structured affective triplet space or free-form descriptions: $Z_{\text{exp}} \in \mathcal{S}_{\text{triplet}} \cup \mathcal{L}_{\text{desc}}$, where $\mathcal{S}_{\text{triplet}} = \mathcal{V}_{\mathcal{R}} \times \mathcal{V}_{\mathcal{I}} \times \mathcal{V}_{\mathcal{E}}$. The synthesis process is modeled as sampling $\hat{A} \sim P(A \mid T, Z_{\text{exp}})$. Unlike turn-based concatenation in standard TTS (Rong et al., 2025; Du et al., 2024), we directly model $A$ as a continuous dialogue speech stream, naturally embedding speaker transitions and overlaps without explicit boundaries markers, similar to *Zero-Shot Dialogue Generation* (ZSDG) (Zhang et al., 2024).

**Challenges.** Compared with conventional *Controllable Text-To-Speech* (CTTS) and ZSDG, our task presents several unique challenges: (i) generating a continuous single-pass end-to-end dialogue audio stream that naturally encodes rich multi-speaker interactions beyond turn-level concatenation; (ii) maintaining coherence and consistency across successive speakers, such as preserving role identity and interactional dynamics throughout multiple turns; and (iii) supporting multi-level controllability, ranging from global conditions specified by structured triplets to fine-grained per-speaker expressive trajectories.

## 4.3 TASK 2: VISION-CONDITIONED DIALOGUE SPEECH SYNTHESIS

**Definition.** This task operates under *implicit controllability*, where the control variable is inferred from visual cues.Let input $\mathcal{C} = \{V_{\text{key}}, T\}$ consist of a temporally ordered keyframe sequence $V_{\text{key}} = \{v_1, \ldots, v_K\}$ and aligned transcripts $T = \{u_1, \ldots, u_K\}$, the goal is to synthesize multi-speaker dialogue speech $A$ that reflects the interaction-level conditions implied by the visual scene. Formally, the model infers a latent style representation $Z_{\text{imp}} = \psi(V_{\text{key}})$ from the visual sequence and generates the audio conditioned on it: $\hat{A} \sim P(A \mid T, Z_{\text{imp}}) = P(A \mid T, \psi(V_{\text{key}}))$.

**Challenges.** Compared with explicit prompt-based control, this task requires the model to (i) reliably infer interactional variables from visual cues such as appearance, posture, and scene composition; (ii) capture temporal dependencies across the keyframe sequence to reflect evolving interactional dynamics in generated speech; and (iii) align inferred styles with textual content $T$ so that the synthesized audio remains both semantically faithful and contextually expressive.

## 4.4 TASK 3: SPEECH-DRIVEN DIALOGUE VIDEO GENERATION

**Definition.** This task instantiates MDG with $\mathcal{C} = \{T, A\}$ and $\mathcal{Y} = \{V\}$. Given dialogue audio $A$ and the corresponding transcript $T$, the objective is to synthesize a dialogue video $\hat{V}$ that is temporally synchronized with speech and affectively consistent with dialogue semantics: $\hat{V} \sim P(V \mid A, T, Z)$, where $Z$ represents the control variables. In explicit control, $Z$ is provided by the user; in implicit control, $Z$ is a latent representation inferred from $A$ and $T$ (prosody, turn-taking, and affective dynamics).

**Challenges.** Compared with text-to-video (T2V) tasks and single talking head generation, our task introduces three key challenges: (i) multi-speaker identity and scene continuity under rapid shot changes and partial visibility; (ii) multi-granularity audio-visual alignment—from lip-audio sync and utterance-level prosody/gesture to dialogue-level expressiveness (relationship, interaction mode, affective state), often under weak/implicit control; and (iii) long-range cinematic reasoning to faithfully stage interactions (who, how, where), requiring shot planning and blocking beyond what standard quality or lip-sync metrics specify.

## 5 BENCHMARKING IN MULTIMODAL DIALOGUE GENERATION

In this section, we first conduct experiments to verify the effectiveness of MM-DIA in supporting the explicit style control in Dialogue Speech Synthesis (Sec. 5.1). We then leverage MM-DIA-

BENCH to reveal key limitations in maintaining audio-visual consistency under implicit cross-modal control in Vision-conditioned Dialogue Speech Synthesis (Sec. 5.2) and Speech-driven Dialogue Video Generation (Sec. 5.3).

## 5.1 EXPERIMENTS ON EXPLICIT CONTROL IN DIALOGUE SPEECH SYNTHESIS

**A. Evaluation Settings.**

*1. Test sets:* We evaluate our model on three evaluation splits: *Hard*, *Test*, and *Out-of-Domain (OOD)*. The *Hard* set is a superset of MM-DIA-BENCH containing 598 clips of highly-expressive data across MM-DIA. The remaining scope of MM-DIA is then randomly split into *Train*, *Valid* and *Test* by 90 : 5 : 5. To further examine generalizability, we additionally construct an *OOD* set with 60 human-refined dialogue clips from daily scenarios. All experimental inference is performed twice, with the Description and Affective Triplet serving as the style control respectively.

*2. Metrics:* To evaluate the performance of the synthesized dialogue speech intrigued by MM-DIA, we established evaluation from the speech-, dialogue-, and controllability-level.

*Speech Quality:* We use word error rate (*WER*) and *UTMOS* (Takaaki et al., 2022) to access the intelligibility and the overall quality of speech.

*Dialogue Quality:* We adopt speaker turn-taking accuracy (*cp-WER*) and speaker aware similarity (*sa-SIM*) respectively to represent the intra-speaker similarity and inter-speaker timbre transition accuracy in spoken dialogue generation.

*Expressiveness Controllability:* Since there are no appropriate objective metric to reflect consistency between the text prompt and speech, we conduct *Human-Mos* evaluation on overall Quality and Instruction-Following capability. The *Human-Mos* score was conducted by 10 participants rating 80 audio samples on a 1-5 scale for both dimensions. Following MoonCast (Ju et al.), we further involve a *Gemini-as-Judge* protocol to enable large-scale nuanced evaluation across Spontaneity, Coherence, Intelligibility, Timbre Similarity, Quality, and Instruction Following. Additionally, we report the Mean Recall Accuracy on the label attributes of Relationship and Interaction Mode.

**B. Baseline Models & Implementation Details.**

To validate the effectiveness of MM-DIA, we perform supervised finetuning of pretrained backbones to enable controllability under both the Triplet and Description control. We select two state-of-the-art pretrained backbones: (1) **Higgs-Audio-V2-Base** (Boson AI, 2025) and (2) **Dia-1.6B** (Nari Labs, 2025). Both models support single-pass dialogue speech generation. Notably, Higgs-Audio-V2 allows flexible conditional inputs across multiple tasks, whereas Dia-1.6B is optimized for dialogue synthesis and lacks native support for conditional input. To enable controllability, we introduce a lightweight adapter module that projects explicit style embeddings into Dia-1.6B's decoder.

**C. Evaluation Results**

Experimental results in Tab. 4 show that Higgs-Audio achieves superior performance in style-controllable dialogue generation after fine-tuning on MM-DIA. The fine-tuning significantly reduces WER ($31.3 \rightarrow 4.5$), and the substantial decrease in cp-WER ($104.8 \rightarrow 33.8$) indicates marked improvement in its original capabilities like content accuracy and dialogue tone conversion. A slight reduction in sa-SIM is observed ($0.48 \rightarrow 0.45$), suggesting a mild trade-off, where improved style controllability is accompanied by slightly reduced speaker timbre consistency, partially due to the increased speaker and style variability in movie-sourced data. While both backbones benefit from fine-tuning on MM-DIA, the stronger performance of Higgs-Audio demonstrates that MM-DIA effectively enhances dialogue accuracy and controllability, particularly when paired with backbones designed for conditional generation. Results under the Affective Triplet control setting, including evaluations on *Hard, Test, and OOD* splits, are provided in Appendix A.5.

## 5.2 EXPERIMENTS ON VISION-CONDITIONED DIALOGUE SPEECH SYNTHESIS

**A. Evaluation Settings**

*1. Test sets:* We use 133 clips from MM-DIA-BENCH, which guarantee single speaker visibility in each keyframe for the model to facilitate accurate speaker attribution. The dialogues exhibit

Table 4: Results of Dialogue Speech Synthesis with **Description** as style control on the ***Test*** set.

| Model | Speech-Quality | | Dialogue-Quality | | Human-MOS | | Gemini-as-Judge | | | | | |
|---|---|---|---|---|---|---|---|---|---|---|---|---|
| | WER↓ | UTMOS↑ | sa-SIM↑ | cp-WER↓ | Qual.↑ | Instr. Follow.↑ | Spont.↑ | Coher.↑ | Intellig.↑ | Similar.↑ | Qual.↑ | Instr. Follow.↑ |
| **Dia-Base** | 19.991 | 2.272 | 0.389 | 51.713 | $2.410_{\pm 0.940}$ | $2.500_{\pm 0.890}$ | 3.993 | 4.335 | 4.446 | 3.738 | 4.248 | 3.807 |
| **Dia-SFT** | 29.071 | 1.974 | 0.447 | 57.813 | $2.890_{\pm 0.690}$ | $2.880_{\pm 0.710}$ | 3.626 | 4.071 | 4.171 | 3.590 | 3.971 | 3.598 |
| **Higgs-Audio-V2-Base** | 31.251 | 3.093 | **0.475** | 104.867 | $3.580_{\pm 0.560}$ | $3.110_{\pm 0.600}$ | 3.313 | 3.96 | 4.276 | 4.021 | 3.874 | 4.012 |
| **Higgs-Audio-V2-SFT** | **4.450** | **3.280** | 0.447 | **33.765** | $4.440_{\pm 0.290}$ | $4.130_{\pm 0.520}$ | **4.277** | **4.881** | **4.965** | **4.640** | **4.851** | **4.707** |

Table 5: Results of Vision-conditioned Dialogue Speech Synthesis on MM-DIA-BENCH.

| Model | Speech-Quality | | Dialogue-Quality | | Label-Recall | Gemini-as-Judge | | | | | |
|---|---|---|---|---|---|---|---|---|---|---|---|
| | WER↓ | UTMOS↑ | sa-SIM↑ | cp-WER↓ | Mean Acc. ↑ | Spont.↑ | Coher.↑ | Intellig.↑ | Similar.↑ | Qual.↑ | Instr. Follow.↑ |
| **HarmoniVox** | 21.223 | 3.5704 | 0.62 | 30.981 | 40.47 | 1.790 | 3.390 | 4.238 | 1.657 | 1.895 | 2.410 |
| **Cascaded Gemini + Higgs** | 5.781 | 3.3245 | 0.499 | 16.267 | 42.33 | 3.081 | 4.129 | 4.927 | 2.605 | 3.21 | 3.347 |
| **Cascaded GPT + Higgs** | 5.793 | 3.4384 | 0.476 | 14.583 | 52.17 | 3.326 | 4.000 | 4.978 | 3.022 | 3.587 | 3.522 |

clear emotional contrasts through visual cues such as facial expressions or body gestures to guide style-consistent speech generation.

***2. Metrics:*** Since Task 2 shares the same output paradigm as Task 1, we preserve most metrics while slightly modifying prompts in Gemini-as-Judge protocol to analyze the alignment in dialogue expressiveness between the speech and visual sequences.

## B. Baseline Models & Implementation Details

We implement several representative baselines for comparison: (1) **HarmoniVox** (Zhou et al., 2025a). The model infers the avatar's latent internal states from a visual image and projects it into a talking style representation to conditioned speech synthesis. We adopt sentence-level inference in our experiments and concatenate corresponding utterances into complete dialogue. (2) **Cascaded VLM + Higgs-Audio-SFT**. We employ a strong vision-language model (e.g., GPT-5, Gemini-2.5-pro) to first generate descriptive style prompts in human interaction from the visual dialogue context, which are then used to condition Higgs-Audio-V2-SFT for speech synthesis.

## C. Evaluation Results

As shown in Tab. 5, the cascaded methods outperform the end-to-end HarmoniVox in most metrics. Comparing with the explicit style-prompt setting in in Tab. 4, although most objective results preserved stable performance related to speech and dialogue quality, the subjective scores in Gemini-as-Judge exhibit a noticeable decline, especially in dimensions like Timbre Similarity and Instruction Following. This suggests that while basic speech synthesis performance is preserved, cross-modal style consistency degrades when cues are provided implicitly through visual interaction context. These results reveal an imbalance in existing approaches: they can produce fluent dialogue, yet struggle to maintain coherent interaction-level style alignment across modalities.

## 5.3 EXPERIMENTS ON SPEECH-DRIVEN DIALOGUE VIDEO GENERATION

### A. Evaluation Settings

***1. Test sets:*** We also use the 133 dialogues from MM-DIA-BENCH, which ensures single-speaker visibility as introduced in Sec. 5.2. This set is curated to cover all annotated relationships and interaction modes in MM-DIA, ensuring broad semantic coverage for cross-modal alignment assessment.

***2. Metrics:*** We evaluate along three axes: *video quality* (Fréchet Video Distance, FVD (Unterthiner et al., 2019)), *lip-speech synchronization* (*LSE-C* and *LSE-D* (Chung & Zisserman, 2016)), and *cross-modal semantics/alignment*. We adopt the model-as-judge pipeline introduced in Sec. 5.1 to score *Spontaneity*, *Coherence*, *Intelligibility*, *Similarity*, *Overall Quality*, and *Instruction Following*, to quantify how well the generated dialogue videos align with the speech modality—from low-level timing (lip-speech sync) and utterance-level prosody/expressiveness to dialogue-level semantics (e.g., staging, flow, and instruction following). In addition, we report label accuracy/recall on *Relationship* and *Interaction Mode* to test whether generated scenes faithfully reflect dialogue-level interpersonal semantics.

Table 6: Results of Speech-driven Dialogue Video Synthesis.

| Model | Visual-Quality | Lip-Sync | | Label-Recall | | Gemini-as-Judge | | | | | |
|---|---|---|---|---|---|---|---|---|---|---|---|
| | FVD↓ | LSE-C↑ | LSE-D↓ | ACC-Rela. ↑ | ACC-Interact.↑ | Spont.↑ | Coher.↑ | Intellig.↑ | Similar.↑ | Qual.↑ | Instr. Follow.↑ |
| FLOAT (Ki et al., 2025) | 572.187 | 4.805 | 9.502 | - | - | 2.703 | 2.405 | 3.050 | 3.339 | 2.248 | 3.050 |
| MultiTalk (Kong et al., 2025b) | 124.543 | **5.305** | 8.795 | - | - | 4.524 | 4.388 | 4.612 | 4.689 | **4.922** | 4.631 |
| Sonic (Ji et al., 2025) | **117.096** | 4.986 | **8.503** | - | - | **4.592** | **4.583** | **4.750** | **4.800** | 4.833 | **4.750** |
| Wan-2.2 S2V (Wan et al., 2025) | 154.261 | 4.288 | 9.873 | - | - | 4.205 | 4.116 | 4.357 | 4.589 | 4.652 | 4.384 |
| HunyuanVideo (Kong et al., 2025a) | 335.591 | - | - | 47.97% | 13.82% | 2.089 | 4.553 | 4.049 | 2.968 | 4.309 | 2.293 |
| Wan-2.2 T2V (Wan et al., 2025) | 300.092 | - | - | **53.66%** | **18.70%** | 3.114 | 4.634 | 4.602 | 3.732 | 4.423 | 3.268 |
| Ground Truth | - | 6.275 | 8.333 | 100.00% | 100.00% | 4.892 | 4.971 | 4.961 | 4.931 | 5.000 | 4.902 |

## B. Baseline Models & Implementation Details

Because no system currently performs end-to-end dialogue-to-video generation, we evaluate two practical families:

- SI2V (Speaker-Image-to-Video). We split dialogue-level movie clips into sentence-level segments and drive the corresponding speaker images with each utterance, then concatenate per-sentence clips into dialogue videos. Given that SI2V models use reference keyframes, we do not evaluate relationship/scene accuracy here; we focus on lip sync and expressiveness alignment.
- T2V (Text-to-Video). Using sentence-level fine-grained and dialogue-level expressiveness annotations in MM-DIA, we construct rich text prompts to condition multi-speaker scene synthesis. Since audio is not explicitly input, we do not score lip sync for T2V; instead, we emphasize relationship/interaction and expressiveness alignment. During model-as-judge, we provide the corresponding audio for Gemini to evaluate the cross-modality alignment.

## C. Evaluation Results

All experiments are conducted on MM-DIA-BENCH dialogue clips with visible dyads and diverse expressiveness to ensure comparable shot complexity across systems.

Results in Tab. 15 show that no current system adequately solves dialogue video generation. Despite rich prompts, T2V models capture only a portion of high-level dialogue semantics; accurate staging of interaction scenes and who-interacts-with-whom remains unreliable. SI2V systems attain higher *Coherence*/*Intelligibility*/*Quality* on average, but *Instruction Following* and fine-grained *Spontaneity* alignment fluctuate across long dialogues.

To summarize, **SI2V** pipelines are complex and depend on keyframes; practical deployment will require coupling with keyframe generation to approach end-to-end usage. Additionally, small face extents and occlusions in natural dialogue shots make lip-sync brittle, often producing artifacts. Meanwhile, **T2V** systems lack explicit audio conditioning, making it difficult to synchronize with speech timing and match vocal expressiveness; they also underperform at faithfully reconstructing relationships and interaction patterns.

Overall, *neither* family is yet adequate for dialogue video generation. The results validate our benchmark design: quality and lip-sync alone are insufficient; cross-modal semantic alignment must be measured explicitly to drive progress. Future work should target: (1) End-to-end dialogue-to-video modeling that unifies keyframe planning, character visibility, lip/body sync, and scene continuity; (2) Multi-granularity alignment learning using sentence-level and dialogue-level expressiveness labels (relationship, interaction mode, affect); (3) Cross-modal semantic discriminators that penalize misalignment during training; and (4) Long-range dependence & shot planning for controllable staging in multi-speaker scenes, consistent with expressiveness schema in MM-DIA.

## 6 CONCLUSION

In this paper, we propose MM-DIA, the first large-scale highly-expressive multimodal dialogue dataset for the task of Multimodal Dialogue Generation, and the corresponding dual-speaker benchmark MM-DIA-BENCH for the evaluation of cross-modal conditional generation tasks. Experiments demonstrate that MM-DIA enhances the style controllability of dialogue generation model and MM-DIA-BENCH reveals the limitation in current cross-modal style consistency.

## 7    ACKNOWLEDGEMENTS

This work is supported by the National Natural Science Foundation of China Nos. 62425604 and 62502256. It is also supported by Beijing Natural Science Foundation (L257006), and High Performance Computing Center, Tsinghua University.

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

## REPRODUCIBILITY STATEMENT

We provide the MM-DIA dataset, a large-scale multimodal dialogue corpus, and the MM-DIA-BENCH benchmark, both of which are integral to our research on style-controllable multimodal dialogue generation. Our experimental code and data curation pipeline will be made publicly available upon acceptance of the paper. The models and algorithms used in this paper can be reproduced using the provided dataset and benchmark, with all necessary details regarding model configurations, training procedures, and evaluation protocols included.

## ETHICS STATEMENT

The MM-DIA and MM-DIA-BENCH datasets include multimodal data sourced from movies and TV series, some of which may contain commercial content. We do not release the video or audio clips themselves; instead, we provide annotations (e.g., transcript, affective triplet, dialogue description, speaker identity, keyframe with active speaker, etc.d) and the methods used to generate them. Researchers are encouraged to obtain the corresponding media content independently and align it with the provided timestamps. For any further queries or information, readers are welcome to contact us.

We acknowledge the potential for biases inherent in the media content used and are committed to addressing these in future versions of the dataset by incorporating more diverse sources and refining our annotation methods.

## LLM USAGE DISCLOSURE

We used GPT-5 for grammar checking and improving the clarity of sections 1 through 6 in this manuscript. All technical content, experimental design, and analysis are original human work. The LLM suggestions were manually reviewed and modified to ensure that they align with the paper's objectives and maintain technical accuracy.

# A APPENDIX

## A.1 IMPLEMENTATION DETAILS FOR SUBTITLE CALIBRATION

We combine multi-sourced uncalibrated subtitles with ASR results to perform precise synchronization between the timestamps and content. Selecting the matched specific ASR segments and subtitle entries as anchor points, we perform translation operations to adjust time and duration differences in the uncalibrated subtitle timestamps with minimal discrepancies. Typical cases in multi-sourced movie subtile alignment are shown in Fig. 4. The qualified subtitle with low variance in discrepancy are double-checked by human to ensure usability. The alignment not only improves the overall synchronization of subtitles with the spoken content, but also mitigates errors introduced by ASR, enhancing the accuracy and reliability of the subtitle data in audio-visual applications.

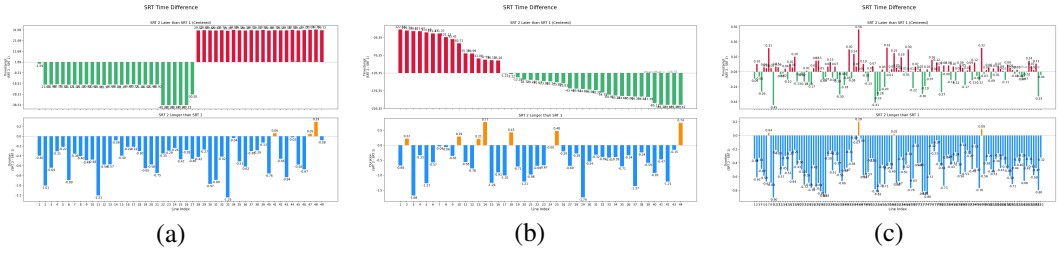

|       (a)       |       (b)       |       (c)       |

Figure 4: Three bad/good cases of subtitle alignment: (a) edited movie segments, (b) edited movie speed, and (c) potential usability with time translation. In each figure, the upper plot shows the start time discrepancy between anchor point start times in the subtitle and the ASR results, and the lower plot shows the duration discrepancy.

## A.2 IMPLEMENTATION DETAILS FOR DIALOGUE EXTRACTION

We introduce a buffer mechanism to dynamically update a keyframe pool of the current scene, with the pseudo-code illustrated in Algorithm 1.

Let $P = \{p_1, p_2, \ldots, p_m\}$ represent the dynamic set of most representative keyframes from the current scene $S = \{s_{t-n}, \ldots, s_{t-1}, s_t\}$, VLM uses the updated keyframes $P$ to perform sparse comparisons of the similarity between the $P$ and the frame after a certain buffer interval $b$, $s_{t+b}$. Whenever the match fails, it falls back to the subsequent frame $s_{t+1}$ through binary search. Once the match is successful, sparse comparisons will start from the new end frame, recognizing the passed frames within the same scene. Meanwhile, the keyframe pool $P$ is updated the by replacing a most similar frame within the pool with the new $s_{t+b}$.

$$S' = \{s_{t-n}, \ldots, s_{t+b}\}, \quad P' = P \cup \{s_{t+b}\} \setminus \{p_{\text{most\_similar}}\}, \quad \text{if} \quad \text{VLM}(P, s_{t+b}) = \text{True}.$$

The buffer spanning multiple frames, together with the memory pool, enables the algorithm to "bridge" temporary interruptions instead of triggering incorrect scene boundaries. This allows the algorithm to maintain the continuity of dialogue scenes over longer periods, providing greater resilience to the complex visual dynamics of movies.

## A.3 VALIDATION RESULTS OF THE ANNOTATION SYSTEM AND DATASET

**1. Evaluation of the correctness in movie-subtitle synchronization.**

As shown in Tab. 7, with the *official* version of subtitle stands for the ground truth of content and human judgment on correctness of timestamps boundaries, the calibrated subtitle performs balanced in low word error with high time accuracy. Notably, both ASR and official subtitle tend to present the line slightly earlier than the actual time, while the start time is usually correct. As a result, we slightly extend the audio up to the next starting time in the subsequent training.

**2. Evaluation on the buffer mechanism in boundary detection.**

---

**Algorithm 1** Subtitle Scene Segmentation with VLM

---

**Require:** Subtitle file srt, Video file video, Step size step, Buffer size buffer
**Ensure:** List of dialogue ranges

 1: Load VLM model (Qwen2.5-VL-7B-Instruct) ParseScriptSrt
 2: Extract subtitle blocks with index and timecode
 3: **return** list of blocks ExtractFramevideo, timecode
 4: Compute midpoint timestamp
 5: Use ffmpeg to extract frame image
 6: **return** image path IsContinuationframes
 7: Prompt VLM with frames to check scene continuity
 8: **if** last frame matches context **then**
 9:     **return** True
10: **else**
11:     **return** False
12: **end if**
13: Initialize ranges list
14: **for** each block i **do**
15:     Try to extend range by comparing future blocks using VLM
16:     Allow up to step ahead, using up to buffer context frames
17:     **if** continuation fails **then**
18:         finalize current segment
19:     **end if**
20: **end for**
21: **return** ranges
22: Save ranges to JSON output

---

Table 7: TimeStamp accuracy and WER of different subtitle version.

| Data Source | TimeStamp Acc. | WER |
|---|---|---|
| ASR | **0.871** | 0.34 |
| SRT-Uncalibrated | 0.179 | 0.43 |
| SRT-Calibrated | 0.857 | 0.03 |
| SRT-Official | 0.870 | **0.00** |

Table 8: Completeness and Hallucination of Dialogue Annotation from Qwen-72B, GPT-5 & Gemini-2.5-pro.

| Annotation | Model | Comp. ↑ | Hall. ↓ |
|---|---|---|---|
| **Non-verbal Sound** | Qwen | 1.25 | 2.12 |
| | GPT | 1.18 | **1.00** |
| | Gemini | **4.66** | 1.22 |
| **Affective Triplet** | Qwen | 3.45 | 2.56 |
| | GPT | 3.66 | 2.20 |
| | Gemini | **4.76** | **1.38** |
| **Description** | Qwen | 3.15 | 2.76 |
| | GPT | 3.60 | 2.16 |
| | Gemini | **4.72** | **1.44** |

Firstly, we conduct human evaluation on a random sampled test set with six movies, with the reported boundary extraction accuracy to be 95.2%, comparing to 86.3% on the traditional frame-by-frame scene continuation detection methods.

As to the ablation study on the proposed buffer mechanism, inspired by the Intersection over Union (IoU) metric commonly used in *Object Detection*, we introduce a new metric called $F1_{Overlap}$ to represent the similarity between two continuous segmentation of a same sequence of clips, expressed as $\{A\}, \{B\}$:

Table 9: Ablation study on the buffer $b$ with Qwen 7B and Qwen 72B model as VLM.

| $F1\_Overlap$ | b=1 | 2 | 3 | 4 | 5 |
|---|---|---|---|---|---|
| Qwen 7B | 0.771 | **0.866** | 0.841 | 0.839 | 0.836 |
| Qwen 72B | 0.947 | 0.975 | 0.977 | 0.978 | **0.979** |

Using $A$ as the reference segmentation, for the $n$ intervals in $A$, we take the corresponding interval in $B$ that has the maximum overlap with it to calculate the percentage of the total overlapping duration of these $n$ overlaps in $A$, denoted as $P(A, B)$. Formally, this can be written as: $P(A, B) = \frac{\sum_{i=1}^{n} \text{Overlap}(A_i, B_{\max})}{\sum_{i=1}^{n} \text{Duration}(A_i)}$. Similarly, we reverse the roles of $A$ and $B$ to compute $P(B, A)$. The similarity between the two segmentation is then computed using the F1 score of $P(A, B)$ and $P(B, A)$: $F1_{Overlap} = 2 \times \frac{P(A,B) \times P(B,A)}{P(A,B) + P(B,A)}$. The $F1_{Overlap}$ metric prevents extreme segments, whether excessively dense or sparse, from receiving a high $P$ score based on a single perspective. As shown in Tab. 9, we leverage Qwen 72B with $b = 3$ to balance the time and performance.

**3. Quality evaluation on the dialogue annotation.**

Following MovieBench Wu et al. (2025), we invite two human annotators to evaluate the performance of Gemini annotation in the data curation pipeline, from the perspective of **Completeness** and **Hallucination**. Annotators are asked to score on a 1-5 scale for the three kinds of annotation of 100 randomly sampled movie/TV clips from MM-DIA. As indicated in Tab. 8, in comparison with Qwen 72B and GPT 5 (which instead takes sequential frames and audio as video input), Gemini-2.5-pro outperforms in most aspects with the best interpretation of the movie style.

### A.4 METRICS EXPLANATION IN TASK 1.

*Speech Quality: **WER**, **UTMOS**.*

We use the official implementation from Zhu et al. (2025) to compute WER and UTMOS, accessing the intelligibility and the overall quality of speech.

*Dialogue Quality: **cp-WER**, **sa-SIM**.*

Speaker Turn-Taking Accuracy (cp-WER) is computed by firstly concatenating all speech utterances by the same speaker after processing the speaker diarization to the generated spoken dialogue, then picking up the lowest WER among all the permutations of the generated transcripts with the concatenated ground truth.

Speaker Aware Similarity (sa-SIM) is acquired by computing mean speaker similarity among the permutations of each speaker's utterance after conducting the Montreal-Forced-Alignment.

### A.5 ADDITIONAL EXPERIMENTAL RESULTS OF DIALOGUE SPEECH SYNTHESIS UNDER AFFECTIVE TRIPLET STYLE CONTROL.

As illustrated in Tab. 10, the results under the Affective Triplet control setting exhibit trends highly consistent with those reported in Tab. 4 under the Description prompt. Across both *Test* and *Hard* splits, Higgs-Audio-V2-SFT consistently achieves the best performance in speech quality, dialogue alignment, and controllability metrics. This consistency indicates that MM-DIA enables robust style control under both natural language and structured affective prompts.

As expected, performance on the *Hard* split (Tab. 11) is generally lower than on the *Test* split (Tab. 10), particularly in WER and coherence-related metrics. This suggests that highly expressive multimodal dialogues introduce increased acoustic variability and emotional volatility, posing additional challenges for speech synthesis. Nevertheless, the relative ranking between models remains stable, and controllability metrics (e.g., label recall and Instruction Following) remain strong, indicating preserved control effectiveness under expressive conditions.

Additionally, as introduced in Sec. 5.1, we manually created *Out-of-Domain* clips of dialogue content and style annotations to enable variable control in dialogue generation. The dialogues are daily conversation that GPT-generated and human-refined. Specifically, data are categorized into every three pieces by the Affective Triplet, with two variables fixed in each group, and the third variable changing. Besides, the three dialogues share a same sentence to be presented in different cases. For instance, the sentence *'Please, listen, if you'd just give me a second, I can clear this up!'* could be acted with subtle difference under different speakers relationships, ranging from *Lovers*, *Employer-employee*, to *Police-Criminal*. Please refer to the audio samples on the Demo Page [2], which clearly

---

[2] https://mmdiaiclr26.github.io/mmdiaiclr26/

demonstrate the model's ability to capture the subtle stylistic shifts corresponding to the controlled variable changes.

Overall, these findings confirm that MM-DIA supports both robust style control and generalization across difficulty levels, while highlighting the intrinsic challenges of expressive dialogue speech synthesis.

Table 10: Results of Dialogue Speech Synthesis with **Affective Triplet** as style control on the ***Test*** set.

| Model | Speech-Quality | | Dialogue-Quality | | Label-Recall | Gemini-as-Judge | | | | | |
|---|---|---|---|---|---|---|---|---|---|---|---|
| | WER↓ | UTMOS↑ | sa-SIM↑ | cp-WER↓ | Mean Acc. ↑ | Spont.↑ | Coher.↑ | Intellig.↑ | Similar.↑ | Qual.↑ | Instr. Follow.↑ |
| **Dia-Base** | 19.991 | 2.272 | 0.389 | 51.713 | 0.210 | 3.452 | 4.000 | 4.161 | 4.016 | 3.887 | 4.113 |
| **Dia-SFT** | 33.178 | 1.941 | 0.430 | 117.947 | 0.237 | 3.636 | 4.118 | 4.187 | 3.910 | 3.962 | 4.014 |
| **Higgs-Audio-V2-Base** | 39.684 | 3.066 | **0.461** | 75.847 | 0.352 | 3.169 | 3.816 | 4.075 | 3.843 | 3.704 | 3.850 |
| **Higgs-Audio-V2-SFT** | **5.265** | **3.286** | 0.459 | **33.134** | **0.428** | **4.031** | **4.820** | **4.967** | **4.610** | **4.636** | **4.809** |

Table 11: Results of Dialogue Speech Synthesis with **Affective Triplet** as style control on the ***Hard*** set.

| Model | Speech-Quality | | Dialogue-Quality | | Label-Recall | Gemini-as-Judge | | | | | |
|---|---|---|---|---|---|---|---|---|---|---|---|
| | WER↓ | UTMOS↑ | sa-SIM↑ | cp-WER↓ | Mean Acc. ↑ | Spont.↑ | Coher.↑ | Intellig.↑ | Similar.↑ | Qual.↑ | Instr. Follow.↑ |
| **Dia-Base** | 20.266 | 2.164 | 0.406 | 71.956 | 0.224 | **4.093** | 4.407 | 4.481 | 3.981 | 4.381 | 4.003 |
| **Dia-SFT** | 28.940 | 1.855 | 0.409 | 68.893 | 0.246 | 3.719 | 4.179 | 4.258 | 4.018 | 4.111 | 4.030 |
| **Higgs-Audio-V2-Base** | 22.041 | **3.711** | **0.463** | 57.850 | 0.327 | 3.190 | 3.969 | 4.637 | 4.151 | 4.115 | 4.095 |
| **Higgs-Audio-V2-SFT** | **7.206** | 3.184 | 0.425 | **35.690** | **0.360** | 3.267 | **4.667** | **4.953** | **4.562** | **4.555** | **4.611** |

## A.6 EXPLANATION OF RELATIONSHIP AND INTERACTION MODE CATEGORIES

Table 12: Explanation of Relationship Categories with Typical Labels

| Relationship | Explanation and Example |
|---|---|
| **Workplace** | Refers to professional relationships and environments, including people within a work setting. Example: *Colleague*, *Boss*, *Manager*, *Coworker*, *Client*. |
| **Friends** | A relationship between individuals characterized by mutual affection, trust, and companionship outside of family and work. Example: *Buddy*, *Pal*, *Companion*, *Mate*, *Peer*. |
| **Intimate** | Relationships of a more personal and romantic nature, typically involving emotional and physical closeness. Example: *Boyfriend*, *Girlfriend*, *Partner*, *Spouse*, *Fiancé*. |
| **Family** | Relationships defined by blood ties or marriage, including extended family members. Example: *Mother*, *Father*, *Sibling*, *Uncle*, *Cousin*. |
| **Adversarial** | Relationships characterized by opposition or conflict, often involving rivalry or animosity. Example: *Enemy*, *Opponent*, *Rival*, *Antagonist*, *Competitor*. |
| **Individual** | A relationship with oneself, or a solitary state where interaction with others is minimal or nonexistent. Example: *Solo*, *Loner*, *Isolated*, *Monologue*. |
| **Social** | Encompasses a wide range of social roles and interactions, from professional settings to casual encounters. Example: *Teacher*, *Doctor*, *Neighbor*, *Stranger*, *Host*, *Customer*. |

| Relationship | Explanation and Example |
|---|---|
| Authority | Relationships based on power and control, typically involving leadership, governance, and decision-making. Example: *King, Judge, Mayor, President, General.* |

Table 13: Explanation of Interaction Mode Categories with Typical Labels

| Interaction Mode | Explanation and Example |
|---|---|
| Suggesting | The act of convincing someone to believe or do something through reasoning or emotional appeal. *Example: Persuasion, Convincing, Negotiation.* |
| Conflict | A state of disagreement or confrontation, often involving tension or hostility. *Example: Argument, Disagreement, Accusation.* |
| Questioning | Asking questions to gain information, clarify doubts, or provoke thought. *Example: Inquiry, Interrogation, Probing.* |
| Narration | The act of narrating a story or personal experience, often to entertain or inform. *Example: Storytelling, Flashback, Monologue.* |
| Explanation | Providing detailed information or clarification on a topic to ensure understanding. *Example: Justification, Diagnosis, Clarification.* |
| Commands | Issuing direct orders or instructions to prompt action. *Example: Orders, Demands, Instruction.* |
| Dynamic Cross-talk | A back-and-forth exchange of dynamic dialogue, often with interruptions or interjections. *Example: Interjection, Interruption.* |
| Sympathizing | Offering comfort or support to someone, often to alleviate concerns or anxiety. *Example: Comfort, Support, Encouragement.* |
| Rejection | Dismissing or refusing a request, idea, or proposal. *Example: Refusal, Dismissal, Avoidance.* |
| Banter | Playful, often teasing, interaction intended to entertain or create rapport. *Example: Teasing, Flirting, Joke.* |
| Authority Power | The use of authority or control to direct others' actions, often in a commanding or corrective manner. *Example: Domination, Criticism, Intervention.* |
| Performance | Delivering a structured or formal presentation, speech, or announcement to an audience. *Example: Presentation, Speech, Announcement.* |
| Reflection | Reflecting on one's thoughts, feelings, or experiences, often leading to a moment of realization. *Example: Introspection, Revelation, Discovery.* |
| Emotion Release | Expressing emotions, often related to frustration, anxiety, or relief. *Example: Venting, Confession.* |
| Invitation | Extending a request for someone to join an event or activity. *Example: Invitation, Offer.* |

Table 14: Comparison of the MM-DIA with dialogue-related datasets across domain, scale, modality, and annotation. Modality includes *text* ($\mathcal{T}$), *vision* ($\mathcal{V}$), and *audio* ($\mathcal{A}$).

| Domain | Dataset | Scale | | | Modality | | | Annotation | |
| | | #Clip | #Utt. | #Dur.(h) | $\mathcal{T}$ | $\mathcal{V}$ | $\mathcal{A}$ | Granularity | Label |
|---|---|---|---|---|---|---|---|---|---|
| Spoken Dialogue | OpenDialogue | 1M | 6.5M | 6.8K | ✓ | ✗ | ✓ | Dialogue | None |
| Textual Dialogue | OpenViDial 2.0 | - | 5.6M | - | ✓ | ✓ | ✗ | Dialogue | None |
| | YTD-18M | 18M | - | - | ✓ | ✓ | ✗ | Dialogue | None |
| Text-to-Video Generation | OpenVid-1M | 1M | - | 2.1K | ✓ | ✓ | ✗ | Scene | Desc. |
| | Captain Cinema | - | 300K | 500.0 | ✓ | ✓ | ✗ | Shot | Desc. |
| Multimodal Dialogue Understanding | MELD | 1.4K | 14K | 13.6 | ✓ | ✓ | ✓ | Sentence | Tag |
| | MC-EIU | 5.0K | 56K | 53.0 | ✓ | ✓ | ✓ | Sentence | Tag |
| Movie Generation | MovieBench | 16.0K | 61K | 69.2 | ✓ | ✓ | ✓ | Shot/Scene | Desc. |
| **Multimodal Dialogue Generation** | **MM-DIA (Ours)** | **54.7K** | **449K** | **360.3** | ✓ | ✓ | ✓ | **Dia./Sent.** | **Desc./Tag** |
| **Multimodal Dialogue Generation** | **MM-DIA-BENCH (Ours)** | **309** | **1,851** | **1.7** | ✓ | ✓ | ✓ | **Dia./Sent.** | **Desc./Tag** |

Table 15: Results of Speech-driven Dialogue Video Synthesis.

| Model | Visual-Quality | Lip-Sync | | Label-Recall | | Gemini-as-Judge | | | | | |
| | FVD↓ | LSE-C↑ | LSE-D↓ | ACC-Rela. ↑ | ACC-Interact.↑ | Spont.↑ | Coher.↑ | Intellig.↑ | Similar.↑ | Qual.↑ | Instr. Follow.↑ |
|---|---|---|---|---|---|---|---|---|---|---|---|
| FLOAT | 572.187 | 4.805 | 9.502 | - | - | 2.703 | 2.405 | 3.050 | 3.339 | 2.248 | 3.050 |
| MultiTalk | 124.543 | **5.305** | 8.795 | - | - | 4.524 | 4.388 | 4.612 | 4.689 | **4.922** | 4.631 |
| Sonic | **117.096** | 4.986 | **8.503** | - | - | **4.592** | **4.583** | **4.750** | **4.800** | 4.833 | **4.750** |
| Wan-2.2 S2V | 154.261 | 4.288 | 9.873 | - | - | 4.205 | 4.116 | 4.357 | 4.589 | 4.652 | 4.384 |
| HunyuanVideo | 335.591 | - | - | 47.97% | 13.82% | 2.089 | 4.553 | 4.049 | 2.968 | 4.309 | 2.293 |
| Wan-2.2 T2V | 300.092 | - | - | **53.66%** | **18.70%** | 3.114 | 4.634 | 4.602 | 3.732 | 4.423 | 3.268 |
| Ground Truth | - | 6.275 | 8.333 | 100.00% | 100.00% | 4.892 | 4.971 | 4.961 | 4.931 | 5.000 | 4.902 |

Table 16: Comparison of the MM-DIA with dialogue-related datasets across domain, source, scale, modality, and annotation.

| Domain | Dataset | Source | Scale | | | Modality | | | Annotation | | |
| | | | #Clip | #Utt. | Dur.(h) | $\mathcal{T}$ | $\mathcal{V}$ | $\mathcal{A}$ | Granularity | Label | Segmentation |
|---|---|---|---|---|---|---|---|---|---|---|---|
| Spoken Dia. Gen. | OpenDialogue | Web | 1M | 6.5M | 6.8K | ✓ | ✗ | ✓ | Dialogue | None | Semantic |
| Textual Dia. Gen. | OpenViDial 2.0 | TV/Mov. | - | 5.6M | - | ✓ | ✓ | ✗ | Dialogue | None | Timestamp |
| | YTD-18M | Web | 18M | - | - | ✓ | ✓ | ✗ | Dialogue | None | - |
| Text-to-Video Gen. | OpenVid-1M | Web | 1M | - | 2.1K | ✓ | ✓ | ✗ | Scene | Desc. | - |
| | Captain Cinema | Movie | - | 300K | 500.0 | ✓ | ✓ | ✗ | Shot | Desc. | Vision |
| Multimodal Dia. Und. | MELD | TV | 1.4K | 14K | 13.6 | ✓ | ✓ | ✓ | Sentence | Tag | Human |
| | MC-EIU | TV | 5.0K | 56K | 53.0 | ✓ | ✓ | ✓ | Sentence | Tag | Human |
| Movie Gen. | MovieBench | Movie | 16.0K | 61K | 69.2 | ✓ | ✓ | ✓ | Shot/Scene | Desc. | Vision |
| Multimodal Dia. Gen. | **MM-DIA** | TV/Mov. | **54.7K** | **449K** | **360.3** | ✓ | ✓ | ✓ | **Dia./Sent.** | **Desc./Tag** | **Multimodal** |
| Multimodal Dia. Gen. | **MM-DIA-BENCH** | TV/Mov. | **309** | **1,851** | **1.7** | ✓ | ✓ | ✓ | **Dia./Sent.** | **Desc./Tag** | **Multimodal** |

