# OpenReview forum: "From Natural Alignment to Conditional Controllability in Multimodal Dialogue"
_ICLR.cc/2026/Conference — ICLR 2026 Poster_

### Official Review · Reviewer_8JZi · 2025-10-30

**Soundness:** 3
**Presentation:** 3
**Contribution:** 3
**Rating:** 6
**Confidence:** 4

**Summary:**

This paper addresses critical limitations in multimodal dialogue generation—specifically the overemphasis on content transmission over style controllability, scarcity of high-quality datasets, and lack of benchmarks for cross-modal consistency. It focuses on achieving expressive, controllable multimodal dialogue through natural alignment of speech, vision, and text, while constructing a large-scale dataset and systematic benchmarks to advance the field.

**Strengths:**

The paper introduces MM-DIA, a dataset curated from 700+ hours of movies/TV series (200+ films, 9 shows) with 360.26 hours of dialogue, 54,700 clips, and 449,138 turns. It features fine-grained annotations across modalities. The paper’s flagship contribution—the MM-DIA dataset—is the first to center on "multimodal dialogue expressiveness". To evaluate implicit cross-modal style consistency (a long-overlooked gap), the paper builds MM-DIA-BENCH—a balanced benchmark of 309 dual-speaker dialogues (1.69 hours, 1,851 turns) with guaranteed speaker visibility. Experiments show MM-DIA significantly enhances style controllability.

**Weaknesses:**

The paper claims MM-DIA and its findings support "a wide range of applications in human–computer interaction, social computing, and film-making" but exclusively uses cinematic data (movies/TV series) for dataset construction and experiments. This creates a critical gap: it is unclear if the proposed framework (annotations, tasks, model insights) generalizes to non-scripted, real-world multimodal dialogue—arguably the most impactful use case for HCI and social computing.

**Questions:**

You claim MM-DIA supports "broad applications in HCI and social computing" but exclusively use cinematic data (movies/TV series) for training and testing. Given that movie dialogue is scripted and emotionally exaggerated (e.g., MM-DIA’s average emotion intensity score of 6.76/10 via Gemini; )—a stark contrast to casual real-world interactions—have you tested if models fine-tuned on MM-DIA (e.g., Higgs-Audio-V2-SFT) retain style controllability on real-world multimodal datasets?

---

> ### Author Response · Authors · 2025-11-23
> **Response to the Reviewer 8JZi**
>
> Thank you for investing your time and expertise in reviewing our work. Your assessment stating that our efforts in implicit cross-modal style consistency as centering on a 'long-over-looked gap' is highly encouraging to us. We are delighted to clarify the concerns and answer the questions you raised.
>
> * *Given that movie dialogue is scripted and emotionally exaggerated (e.g., MM-DIA’s average emotion intensity score of 6.76/10 via Gemini; )—a stark contrast to casual real-world interactions—have you tested if models fine-tuned on MM-DIA (e.g., Higgs-Audio-V2-SFT) retain style controllability on real-world multimodal datasets?*
>
> We thank the reviewer for this important critique, and appreciate you pointing to our emotion intensity metric as evidence for this contrast.
> We would like to clarify that, cinematic dialogue remains rooted in reality. Performances strive for naturalism, where actors frequently employ improvisation to replicate the spontaneous nuances of authentic human interaction.
> Therefore, although some exceptions exist, such as exaggerated expression in comedy or surreal scenes in science fiction, the MM-DIA dataset is not dominated by high-intensity and emotionally exaggerated dialogue. Instead, it consists of a diverse range of emotions, scenarios (e.g., hospitals, workplace, home), relationships (e.g., friends, partners, family), and interaction types (persuasion, quarreling, comforting) that are similar to in real-world scenarios.
>
> To directly address your concern, **we tested our Higgs-Audio-V2-SFT model on a standard real-world spontaneous dataset, please refer to the experiment introduced in the above general response**.
> The results confirm that our model indeed retains strong performance and style controllability on this non-scripted, real-world dataset. This evidence helps bridge the gap in real-world dialogue and scripted acting from movie/TV, demonstrating that the **representations learned from MM-DIA are not confined to movie/TV contexts but are broadly applicable**. We appreciate the reviewer for prompting this important clarification and additional experiment.
>
> **[Final Remark]** Thank you again for investing the time and effort to review our paper and for the helpful comments that helped us improve the submission. We hope that our responses will address your concern on generalizability and sincerely invite you to engage with us if you have more questions.

---

> > ### Comment · Reviewer_8JZi · 2025-11-26
> >
> > I thank the authors for their detailed response. I will maintain my original rating.

---

> > > ### Author Response · Authors · 2025-11-28
> > > **Thanks for your acknowledgement!**
> > >
> > > Thank you very much for acknowledging our rebuttal! We sincerely appreciate the time and thoughtful consideration you devoted to reviewing our work and offering such constructive feedback. We’re pleased that our clarifications helped address your concerns adequately, and we deeply value your positive assessment.

---

### Official Review · Reviewer_KcRL · 2025-10-30

**Soundness:** 3
**Presentation:** 2
**Contribution:** 3
**Rating:** 6
**Confidence:** 3

**Summary:**

The paper introduces MM-DIA, a large-scale, richly annotated multimodal dialogue dataset from movies and TV series, and MM-DIA-BENCH, a benchmark for evaluating cross-modal conditional generation. Experiments show MM-DIA improves style-controllable dialogue generation

**Strengths:**

1. This is the first dataset to focus on dialogue expressiveness across multiple modalities. The benchmark (MM-DIA-BENCH) fills a gap for evaluating cross-modal style consistency, which is underexplored in prior work.
2. The paper provides a unified framework for MDG, with well-defined tasks and evaluation metrics. Experiments are thorough, with both objective and subjective metrics.

**Weaknesses:**

1. The paper’s main contribution is dataset and benchmark creation; the modeling advances are limited to fine-tuning existing architectures and adapter modules for controllability. No novel end-to-end model for multimodal dialogue generation is proposed or evaluated.
2. The paper is too dense and at times it is difficult to follow, especially in the technical details of the pipeline and annotation process.
3. The dataset is sourced primarily from movies and TV series, which may limit the diversity and generalizability to real-world, spontaneous dialogues

**Questions:**

1. How do you envision MM-DIA supporting research on more spontaneous, real-world dialogues?

2. How does your approach handle long-range dependencies, such as multi-turn conversations or scenes with complex speaker dynamics?

---

> ### Author Response · Authors · 2025-11-23
> **Response to the Reviewer KcRL (Part 1/2)**
>
> Thank you for your insightful feedback. We are grateful for your recognition of our motivation, design of evaluation framework, and our pioneer efforts in multimodal dialogue expressiveness. It's our pleasure to answer the questions you've raised.
>
> * *The dataset is sourced primarily from movies and TV series, which may limit the diversity and generalizability to real-world, spontaneous dialogues. ... How do you envision MM-DIA supporting research on more spontaneous, real-world dialogues?*
>
> Thank you for this thoughtful question. While natural, spontaneous conversations are indeed an important aspect of dialogue research, we would like to clarify that our research focuses on a different objective. **Instead of modeling unconstrained, free-form interactions, we aim to enable expressive and controllable dialogue generation within well-defined scenarios**, where context, speaker intent, and stylistic attributes are precisely specified. From this perspective, movie/TV data provides authentic, lifelike conversations with clearly defined real-world settings and character relationships, which are difficult to widely collect from "in the wild" data.
>
> Despite our different focus, MM-Dia still retains naturalistic characteristics of real-world interactions. For example, non-verbal tags such as *laughter*, *sigh*, *chuckles*, and *gasp* appear among the top 20 most frequent tags. This indicates that MM-DIA captures authentic conversational cues and **complements research on spontaneous dialogue**.
>
> To directly verify the generalization to spontaneous scenarios, we conducted the test **on a real-world, spontaneous dialogue dataset, as reported in the experiment in the above General Response**. The results demonstrate that models trained on MM-DIA not only excel at expressive tasks but also **maintain strong performance on general spontaneous dialogue**, demonstrating the broader applicability of our dataset.
>
> * *How does your approach handle long-range dependencies, such as multi-turn conversations or scenes with complex speaker dynamics?*
>
> This is an excellent question that gets to the core challenges of generating realistic dialogue. Our approach addresses these challenges from both a data and a modeling perspective.
>
> 1. **Annotation phase**
>
> To capture long-range dependencies: We preserve long and complete dialogue clips (rather than isolated snippets) through tolerance-enhanced scene boundary detection (Appendix A.1).
> Additionally, the *Affective Triplet* is applied at dialogue-level to enable global context that captures the long-range stylistic dependencies.
>
> To capture complex speaker dynamics: The *Freestyle Description* explicitly describes the evolving emotional flow, turn-taking irregularities, and other complex interactions (e.g., *"they talk over each other until the conversation erupts into laughter"*). Additionally, sentence-level non-verbal annotations (e.g., *interrupt*, *sobbing*) further retains fine-grained speaker dynamics.
>
> 2. **Generation phase**
>
> The model receives the entire dialogue-level style description as **global conditioning**. The prompt guides throughout the entire generation process to maintain stylistic consistency. Compared to iterative turn-level conditional dialogue generation [1,2], we employ **single-pass generation** for the entire dialogue. This allows the model to implicitly learn turn-taking, interruptions, and consistency from the global context, which is the essence of complex dynamics.
>
> 3. **Result**
>
> Resultantly, the SFT model excels in the *cp-WER* metric (SFT-33.8 v.s. Base-104.9, Tab.4), indicating accurate speaker dynamics in multi-turn conversations (4-8 turns). It also receives higher score in dialogue *Coherence* and *Instruction Following*. However, achieving explicit, fine-grained control and developing specific evaluation metrics for such dynamics remain key directions for future research.
>
> Ref:
>
> [1] Liu R, Hu Y, Ren Y, et al. Generative Expressive Conversational Speech Synthesis[C]. arXiv preprint arXiv:2407.21491, 2024.
>
> [2] Jia Z, Liu R. Intra- and Inter-modal Context Interaction Modeling for Conversational Speech Synthesis[C]. arXiv preprint arXiv:2412.18733, 2024.

---

> > ### Author Response · Authors · 2025-11-23
> > **Response to the Reviewer KcRL (Part 2/2)**
> >
> > * *The paper’s main contribution is dataset and benchmark creation; the modeling advances are limited to fine-tuning existing architectures and adapter modules for controllability. No novel end-to-end model for multimodal dialogue generation is proposed or evaluated.*
> >
> > We thank the reviewer for this insight. We prioritized addressing the foundational data gap over architectural novelty, since the lack of large-scale, richly annotated resource for controllable multimodal dialogue made it difficult to properly train or systematically evaluate models for this complex task. We believe that the creation of such a resource will pave the way for substantial methodological progress.
> > Furthermore, our benchmark reveals clear limitations of current SOTA models in dynamic cross-modal consistency, pointing to concrete directions for future architectural improvements.
> > Therefore, we see our paper as a catalyst for this next exciting phase of research. Future work may focus on the joint generative modeling across modalities or more fine-grained conditioning mechanisms.
> >
> > * *The paper is too dense and at times it is difficult to follow, especially in the technical details of the pipeline and annotation process.*
> >
> > We sincerely thank the reviewer for this crucial feedback on the paper's clarity. We recognize that in our effort to be comprehensive, some sections became overly dense. To clarify the workflow immediately, we provide a streamlined summary of our pipeline, which operates in four hierarchical stages:
> >
> > 1. **Subtitle Calibration** \
> > Challenge: Raw subtitles suffer from timestamp drift, while ASR transcripts have accurate timing but high word error rates. \
> > Solution: We implement a hybrid synchronization strategy to align the high-quality text of official subtitles with the precise timestamps of ASR, ensuring textually accurate and temporally synchronized data.
> > 2. **Dialogue Extraction** \
> > Challenge: Traditional scene detection cuts at every scene shift, fragmenting continuous dialogues. \
> > Solution: We employ a "tolerance-enhanced" boundary detection with a buffer mechanism. This allows the model to "tolerate" momentary visual interruptions (e.g., reaction shots or flashbacks) and bridge gaps, preserving the integrity of long, continuous dialogue scenes.
> > 3. **Fine-Grained Annotation (Sentence-Level)** \
> > Challenge: Accurate speaker attribution is complicated due to frequent audio-visual asynchrony in movie/TV (e.g., off-screen speech). \
> > Solution: We employ a multimodal contextual inference approach. We use Gemini-2.5 to reason jointly over the audio-visual streams, context, and character banks, while simultaneously capturing fine-grained sentence-level speaker dynamics.
> > 4. **Expressiveness Annotation (Dialogue-Level)** \
> > Challenge: High-level "dialogue expressiveness" remains under-defined in current literature. \
> > Solution: We propose a novel systematic schema to formalize expressiveness. It comprises a structured Affective Triplets (Relationship, Interaction Type, Emotional State) and a Freestyle Descriptions (per-speaker, turn-level) to capture the dynamics of the conversation.
> >
> > We hope this summary clarifies the logical progression and necessity of each step in our data construction process. We will revise Section 3 to make our method much easier to follow. Thank the reviewer again for this feedback.
> >
> > **[Final Remark]** Thank you once again for recognizing our work and offering invaluable suggestions for improvement. Please feel invited to leave more comments in case you have additional questions.

---

### Official Review · Reviewer_jqoy · 2025-11-01

**Soundness:** 3
**Presentation:** 2
**Contribution:** 3
**Rating:** 4
**Confidence:** 2

**Summary:**

This paper addresses the limitations in scale, expressiveness, and benchmarking of existing datasets for multimodal dialogue generation by proposing a novel data curation and annotation pipeline, resulting in the large-scale and expressive multimodal dialogue dataset MM-DIA. The authors further introduce a unified framework for Multimodal Dialogue Generation (MDG) and define three representative downstream tasks, including style-controllable speech synthesis, vision-conditioned speech synthesis, and speech-driven video generation. Through systematic benchmarks and experiments, the paper demonstrates the effectiveness of the new dataset in enhancing dialogue style controllability and cross-modal consistency, while revealing the shortcomings of current methods in expressiveness and multimodal alignment. This work provides valuable resources and new challenges for future research in conditional multimodal dialogue generation.

**Strengths:**

1. The unified framework for multimodal dialogue generation proposed in this paper is highly practical and extensible, providing systematic task definitions and benchmarking foundations that will facilitate further advances in the field.
2. The experimental section is comprehensive, covering various downstream tasks and systematically benchmarking the dataset and methods for style controllability and cross-modal consistency, with convincing results.
3. The paper is well-structured and clearly articulated, progressing logically from problem motivation, dataset construction, method design, to experimental evaluation, making it easy for readers to follow and understand the research.

**Weaknesses:**

1. Although the dataset is large and expressive, it is mainly sourced from movies and TV series, which may differ from real-life conversations and affect the generalizability of models to real-world scenarios.
2. The evaluation methodology in the paper is insufficient for assessing the generalization ability of the trained models. In Section 5.1, the authors mention an out-of-domain dataset, but do not clearly specify whether it comes from different data sources or demonstrate its differences. The model’s performance in real-world scenarios, as well as the potential degradation of its original capabilities after SFT (fine-tuning), require further consideration and analysis.
3. There are some typographical errors in the paper, such as the table on page 782 not being cited.

**Questions:**

1. In the evaluation, the authors use Gemini as a judge. Has the accuracy of using large models as judges been tested, and has the model’s performance been compared with human evaluation?
2. In this paper, the authors achieve "From Natural Alignment to Conditional Controllability" from a data perspective. From a methodological standpoint, do you think further improvements at the model level could help achieve this goal?

---

> ### Author Response · Authors · 2025-11-23
> **Response to the Reviewer jqoy (Part 1/2)**
>
> We appreciate your careful review and constructive points. We are delighted to note your recognition of our framework, writing and the experimental design. We are glad to answer the questions you've raised.
>
> * *W1: Although the dataset is large and expressive, it is mainly sourced from movies and TV series, which may differ from real-life conversations and affect the generalizability of models to real-world scenarios. W2-2: The model’s performance in real-world scenarios ... require further consideration and analysis.*
>
> Thank you for the valuable insight.
> Since our goal is also to mimic expressive and controllable dialogue in specific real-world scenarios, movie/TV data has an advantage in providing authentic conversations with clearly defined lifelike settings and character relationships.
> The MM-DIA dataset is featured by diverse range of scenarios (hospitals, workplace, home), relationships (friends, partners, family), and interaction types (persuasion, interrogation, comforting). Its width and richness is practically infeasible to capture in a single real-world dataset due to prohibitive costs and significant privacy concerns.
>
> To directly address the question of generalizability, we added a **new experiment on a real-world dialogue dataset**. We respectfully refer our reviewer to **the above general responce** for detailed experimental results, which demonstrate that our trained model **maintains its robust performance on general, everyday conversations.**
>
> * *W2-1: In Section 5.1, the authors mention an out-of-domain dataset, but do not clearly specify whether it comes from different data sources or demonstrate its differences.*
>
> We thank the reviewer for pointing out this clarification issue.
> As introduced in Line 396, we manually created 60 clips of dialogue content and style annotations to enable variable control in dialogue generation. The dialogues are daily conversation that GPT-generated and human-refined.
> The audio samples are demonstrated **on the Demo Page** as '*Task 1-2. Out-of-domain dialogue speech synthesis with Affectie Triplet as style prompt, **with Variable Control**.*'
> Specifically, data are categorized into every three pieces by the Affective Triplet, with two variables fixed in each group, and the third variable changing. Besides, the three dialogues share a same sentence to be presented in different cases. For instance, the sentence *'Please, listen, if you’d just give me a second, I can clear this up!'* could be acted with subtle difference under different speakers relationships, ranging from *Lovers*, *Employer-employee*, to *Police-Criminal*.
> We encourage our reviewer to listen to these audio samples, as we believe they clearly demonstrate the model's ability to capture the subtle stylistic shifts corresponding to these controlled variable changes.
>
> * *W2-3: The evaluation methodology in the paper is insufficient for assessing the generalization ability of the trained models, ... the potential degradation of its original capabilities after SFT (fine-tuning), require further consideration and analysis.*
>
> We appreciate the reviewer's suggestion. To detect whether SFT model presents degradation of Higgs-audio's original capabilities, we reassess the performance of SFT model on three benchmarks (Seed-TTS Eval, ESD, EmergentTTS-Eval) that Higgs-audio officially used.
>
> | | | **Higgs-Base** | **Higgs-SFT** |
> | :--- | :--- | :---: | :---: |
> | **SeedTTS-Eval dataset** | WER↓ | 2.331 | **1.888** |
> | | SIM↑ | 0.625 | **0.632** |
> | **ESD dataset** | EMO-SIM↑ | **0.732** | 0.715 |
> | **EmergentTTS** | WIN Rate↑ | **73.33%** | 72.17% |
>
> After fine-tuning on the MM-Dia dataset, the Higgs v2 model maintains or surpasses the performance of the base model on standard benchmarks for speech synthesis, voice cloning, emotion similarity, and intonation naturalness.
> The result highlights the quality and compatibility of the MM-Dia dataset to enhance the model's controllability for complex dialogue scenarios **while well-behaved in its core synthesis quality**.

---

> ### Author Response · Authors · 2025-11-23
> **Response to the Reviewer jqoy (Part 2/2)**
>
> * *W3: There are some typographical errors in the paper, such as the table on page 782 not being cited.*
>
> We apologize for the typos and will carefully adjust them in the revised version.
>
> * *Q1: In the evaluation, the authors use Gemini as a judge. Has the accuracy of using large models as judges been tested, and has the model’s performance been compared with human evaluation?*
>
> Thank you for the question. We included a **Human-MOS evaluation** in our study, as detailed in Table 4. Specifically, we invited 10 participants to rate 80 audio clips on a 1-5 scale. The results confirm that our SFT model achieves the highest scores in both Audio Quality (SFT 4.44 vs. Base 3.58) and Instruction Following (SFT 4.13 vs. Base 3.11).
> To further quantify the agreement between Gemini and human ratings, we calculated both model-level and clip-level Pearson correlation coefficients $r$. Higher $r$ values indicate stronger agreement. The results are as follows:
>
> | | **Model-level Pearson's r** | **Clip-level Pearson's r** | **Gemini scoring STD** | **Human scoring STD** |
> | :--- | :---: | :---: | :---: | :---: |
> | **Audio Quality** | 0.92 | 0.79 | 0.52 | 0.62 |
> | **Audio Instruction Following** | 0.99 | 0.75 | 0.61 | 0.68 |
>
> The model-level Pearson's $r$ are high, indicating that Gemini's overall assessment of different models strongly aligns with human judgment. The correlation at single audio clip-level is relatively low, but still showing a strong positive correlation (> 0.75), validating Gemini's effectiveness as a proxy for evaluating individual audio samples.
> Regarding the variance, the standard deviation of human scores is slightly higher than Gemini's. This reflects the expected inter-human variability inherent in subjective perception of audio nuances. In contrast, Gemini provides more consistent and reproducible scores, which is a key advantage for scalable evaluation.
>
> * *Q2: In this paper, the authors achieve "From Natural Alignment to Conditional Controllability" from a data perspective. From a methodological standpoint, do you think further improvements at the model level could help achieve this goal?*
>
> This is a thoughtful question. We believe that the primary bottleneck in this field has long been the lack of high-quality, interaction-rich data, which motivated our focus on MM-Dia.
> However, we acknowledge that with such data serves as necessary foundation, further innovations at the model level are critical to fully realize the goal of controllable, naturally aligned multimodal dialogue. To better capture the inherently "natural alignment", future work might move from pipeline-based approaches towards models that enable joint modeling across modalities (e.g., speech and facial expressions), that is, to model the distribution P(audio, video | text, style). This would not only ensure tighter audio-visual coherence, but also allow for more complex conditioning.
>
>
> **[Final Remark]** Thank you again for reviewing our work and for insightful suggestions. We hope that our response addresses your confusions and concerns. Please feel invited to engage with us if you have more questions. We would really appreciate it if you could consider raising the score.

---

### Official Review · Reviewer_Nvzm · 2025-11-01

**Soundness:** 3
**Presentation:** 3
**Contribution:** 3
**Rating:** 6
**Confidence:** 4

**Summary:**

This paper introduces MM-DIA, the first large-scale and highly expressive multimodal dialogue dataset designed for Multimodal Dialogue Generation (MDG). In addition, the authors present MM-DIA-BENCH, a dual-speaker benchmark specifically developed for evaluating cross-modal conditional generation.
Experiments show that training on MM-DIA significantly enhances controllable dialogue generation, while evaluations on MM-DIA-BENCH reveal notable limitations of current models in achieving consistent multimodal style alignment.

**Strengths:**

* This paper is well-motivated, and the proposed dataset paves the way for future research on style controllability of multi-modal dialogue generation.

* Strong experimental validation with ablations on both controllability and user satisfaction metrics.

* This paper is clearly structured and easy to follow.

* Although the dataset creation heavily relies on models, the authors try to demonstrate that the proposed pipeline achieves human-level quality in annotation consistency and reliability

**Weaknesses:**

* The data creation and evaluation partly rely on GPT-based scoring, which could cause an upper bound of future research.

**Questions:**

Pls refer to weaknesses.

---

> ### Author Response · Authors · 2025-11-23
> **Response to the Reviewer Nvzm**
>
> Thank you for your insightful feedback, and for acknowledging our motivation, writing and the design of evaluation framework. It's our pleasure to answer the questions you've raised.
>
> * *The data creation and evaluation partly rely on GPT-based scoring, which could cause an upper bound of future research.*
>
> We acknowledge that discrepancies still exist between the SOTA model outputs and human annotations.
> However, due to the prohibitive cost of large-scale manual annotation, partially leveraging automated methods is currently inevitable.
> To approach human-level quality and mitigate this issue, we made great efforts to ensure the reliability and robustness of the data and metrics from two aspects:
>
> - **Quality control to avoid model artifacts**:
>
> Throughout the data generation process, we implemented multi-stage filtering and verification, supplemented by manual spot-checks to prevent model-induced artifacts or biases (as introduced in Appendix A.2). As shown in Tab.7, human evaluations confirm the accuracy and consistency of Gemini-involved annotations across diverse tasks, from fine-grained non-verbal sound recognition to detailed dialogue style descriptions.
>
> - **Quantitative alignment with human judgment**:
>
> We systematically compared automated scores with human expert ratings. During annotation, the Pearson correlation coefficients ($r$) between Gemini and human scores for $Emotion Intensity$ and $Emotion Flow Volatility$ are 0.84 and 0.87 (Tab.3). During evaluation, system-level Pearson's $r$ for $Quality$ and $Instruction Following$ scores are 0.92 and 0.99 (Tab.4). These consistently high correlations demonstrate that model-based scoring is a reliable proxy for human assessment.
>
>
> In summary, while we acknowledge the current data limitations due to the trade-off between scale and perfect human quality, our comprehensive validation demonstrates **its effectiveness and reliability at this stage for a reasonable performance**. We believe that it paves the way for future research rather than setting a permanent limit.
>
> **[Final Remark]** Thank you again for reviewing our work and for insightful suggestions. We hope that our response addresses your concerns. Please feel invited to engage with us if you have more questions.

---

### Author Response · Authors · 2025-11-23
**General Response**

We sincerely thank our reviewers for their constructive feedback, which has significantly helped to strengthen our manuscript. In this general response, we prioritize addressing a primary concern shared by three reviewers regarding the **generalization to real-world scenarios**, while also clarifying our **research scope** and **methodological contributions**.

**1. Generalization to real-world scenarios**

To evaluate the model's performance in real-world scenarios, we adopted real-world spontaneous dialogue from the commonly-used Switchboard dataset [1] and employed Gemini-2.5 Pro for dialogue style generation. The dataset was filtered to include 232 dual-speaker dialogues (4-8 turns) featuring colloquialisms such as "um-hum" and repetitions. We reconducted the experiment using these dialogues as Tab.4:

| | | **Higgs-Base** | **Higgs-SFT** | **Higgs-Base** | **Higgs-SFT** |
| :--- | :--- | :---: | :---: | :---: | :---: |
| **Control** | | w/o prompt | w/o prompt | Description | Description |
| **TTS-Quality** | WER↓ | 36.133 | 12.087 | 32.598 | **9.496** |
| | UTMOS↑ | **3.4703** | 3.1082 | 3.4232 | 3.2671 |
| **Dialogue-Quality**| sa-SIM↑ | 0.42 | 0.438 | 0.418 | **0.460** |
| | cp-WER↓ | 62.822 | 22.89 | 52.215 | **22.123** |
| **Gemini-as-judge** | Spontaneity↑ | 3.831 | 3.943 | 3.747 | **4.126** |
| | Coherence↑ | 4.13 | 4.443 | 3.93 | **4.590** |
| | Intelligibility↑ | 4.316 | 4.761 | 4.11 | **4.834** |
| | Similarity↑ | 4.048 | 4.07 | 3.939 | **4.166** |
| | Quality↑ | 3.914 | 4.165 | 3.772 | **4.226** |
| | Instruction Following↑ | - | - | 4.013 | **4.563** |

Experimental results demonstrate that the SFT model consistently outperforms the Base model across both control strategies, achieving superior TTS and dialogue quality, which is further validated by MLLM judgments. While the domain gap between MM-Dia and spontaneous speech caused a slight increase in WER compared to Tab.4 (rising from 3.093 to 9.496), the SFT model still maintains a significantly lower error rate than the Base model (32.598). Furthermore, other metrics remain comparable to, or even exceed, those obtained on movie/TV data in Tab.4. These results highlight **the generalization capability and stability of the MM-Dia-finetuned model on out-of-domain data**. It also suggests that the naturalness and emotional coherence from movie/TV data are **helpful in replicating real-life daily conversations**.

**2. Clarification of scope and contributions**

Based on the feedback, we would like to briefly reiterate two key aspects of our work:

(1)   **Distinction from spontaneous dialogue generation:** Unlike research on spontaneous conversation (e.g., ZipVoice-Dialog [2], Mooncast [3]) which prioritize unconditional naturalness and non-verbal vocalizations with less emphasis on control over specific stylistic constraints, **we aim at expressive, conditionally controllable dialogue generation**. We focus on mimicking authentic interactions with precise control over context and style (e.g., relationships, emotional flow), addressing the gap in controllable dialogue generation.

(2)   **Contributions beyond dataset:** While MM-Dia is a core contribution, our work also establish a comprehensive **methodological framework** for the community. This includes:

* A **novel annotation pipeline** that preserves long-range dialogue dependencies and specifies abstract dialogue expressiveness (Affective Triplets, Freestyle Descriptions);

* A **new benchmark (MM-Dia-Bench)**, which reveals the limitations of current SOTA models in cross-modal consistency;

* A **unified Multimodal Dialogue Generation (MDG) formulation** with standardized tasks and metrics to enable future methodological innovations in this field.

We thank our reviewers again for their invaluable time and insights. Detailed responses to specific questions by each reviewer are provided below. We appreciate your reconsideration and hope that our response have fully addressed the reviewers' comments.

Ref:

[1] Godfrey J J, Holliman E C, McDaniel J. SWITCHBOARD: telephone speech corpus for research and development[C]. ICASSP, 1992.

[2] Zhu H, Kang W, Guo L, et al. ZipVoice-Dialog: Non-Autoregressive Spoken Dialogue Generation with Flow Matching[C]. arXiv preprint arXiv:2507.09318, 2025.

[3] Ju Z, Yang D, Yu J, et al. MoonCast: High-Quality Zero-Shot Podcast Generation[C]. arXiv preprint arXiv:2503.14345, 2025.

---

### Author Response · Authors · 2025-11-28
**Gentle Reminder on Rebuttal Review**

Dear Reviewers,

We hope you are doing well. We would like to kindly follow up and ask whether you could take a moment to review our rebuttal and share any updated assessments when convenient. Your time and effort are greatly appreciated. Please let us know if any further clarification would be helpful.

Best regards,

Authors of Submission 24632

---

### Author Response · Authors · 2025-12-03
**Note for the Area Chair**

As the discussion period concludes, we would like to provide a brief summary of the reviews and prior discussions to assist your final assessment. This paper presents MM-DIA and MM-DIA-BENCH, the first large-scale dataset and benchmark specifically designed for **expressive, controllable multimodal dialogue**. All the reviewers recognized it as a significant contribution that fills an important gap in this area.

We extend our heartfelt thanks to Reviewers Nvzm (Score:6, Conf:4), KcRL (Score:6, Conf:3), and 8JZi (Score:6, Conf:4) for their unanimously support for the paper.
We have resolved their technical queries regarding resource limitations, research scope and pipeline clarity (with Reviewer 8JZi explicitly appreciating our detailed response).

We also thank Reviewer jqoy (Score:4, Conf:2) for recognizing our systematic task definition, articulated writing, extensible framework, and comprehensive experimental design.
We have thoroughly addressed jqoy's primary concerns regarding real-world generalization, potential capability degradation, and evaluation rigor through new experiments and validation studies, although we have not yet received a further response.

We provide a summary of the key clarifications and improvements made during the rebuttal as follows:

1. **Regarding Generalization** (KcRL, 8JZi, jqoy): To address the main concern about whether cinematic data generalizes to real-world scenarios, we conducted new experiments on the Switchboard dataset (a real-world spontaneous speech). The fine-tuned model maintains robust performance and style controllability on out-of-domain data, empirically confirming that the **naturalness and emotional coherence inherent in MM-DIA are effectively transferable to real-life daily conversations.**

2. **Regarding Reliability of Evaluation** (Nvzm, jqoy): Addressing concerns on LLM-based metrics, we provided a new Human-MOS evaluation showing **high Pearson correlation** between Gemini scores and human judgment. This validates our evaluation pipeline as a reliable proxy.

3. **Regarding Contribution** (KcRL): We clarified that we prioritized our core contribution in establishing the necessary **infrastructure** (Dataset, Pipeline, Benchmark, and Task Formulation), which truly enables and *"paves the way for future research on style controllability of multi-modal dialogue generation" (Nvzm)*.

**[Final Remark]** We thank the Area Chair for your time and consideration. We appreciate all the reviewers' efforts in helping us refine the manuscript.
The additional experiments and validation studies during the rebuttal period will be integrated into our revision.
Given the strong consensus on the dataset's novelty and the rigorous empirical verification, we believe this work is well-positioned for inclusion in ICLR.


Best regards,

The Authors

---

### Meta-Review · Area_Chair_NZaW · 2026-01-07

**Summary:**

This paper proposes MM-DIA, a large-scale multimodal dialogue dataset curated from movies/TV, and MM-DIA-BENCH, a benchmark aimed at evaluating implicit cross-modal style consistency and controllable multimodal dialogue generation. The paper also provides an MDG task formulation and reports comprehensive experiments showing that training on MM-DIA improves controllability, while MM-DIA-BENCH exposes limitations of current methods in multimodal style alignment.

**Reviewer Concerns:**

This paper has borderline initial opinions (three boderline accepts and one borderline reject). As strengths of this paper, the reviewers mentioned the scale and benchmark design, the novelty of the proposed cross-modal style consistency problem, and the convincing experimental results.

There were several concerns raised by the reviewers.

- Generalization beyond cinematic (scripted) dialogue to non-scripted, real-world interactions (KcRL, 8JZi, jqoy).
    - The rebuttal comment showed additional experiments on Swithboard, a real-world speech, and argued that the fine-tuned model maintains robust performance and style controllability on Switchboard.
- The evaluation heavily relies on LLMs (Nvzm)
    - The rebuttal comment added the human MOS study and reported high Pearson correlations between Gemini and human ratings
- Potential capability degradation after SFT (jqoy)
    - Additional results on SeedTTS-Eval, ESD, and EmergentTTS-Eval showed that SFT largely maintains or improves core capabilities
- Limited novelty on modeling (KcRL)
    - The rebuttal comment clarified that the position of this paper lies on an infrastructure contribution rather than a modeling advance

Overall, I think that this paper has some limitations and weaknesses, but most of the significant concerns were resolved by the rebuttal.
Given the rebuttal (added Switchboard experiment, human validation for LLM-based metrics, degradation checks, and clarified pipeline), I would lean towards acceptance.

**Reviewer Scores:**

I think that the reviewers who gave borderline acceptances would not bump down their opinions below the borderline. They would change their opinions to more positive ones, but I don't think their opinions would be very strong. I think that the reviewer who gave borderline rejection would change their opinion to borderline acceptance, given that their main concerns were the generalizability and the evaluation protocol (LLM dependency), and the rebuttal comment partially addressed the concern in my opinion (by additional experiments and showing MoS results).

---

### Decision · Program_Chairs · 2026-01-26

Accept (Poster)